# Root stem cell homeostasis in *Arabidopsis* involves cell-type specific transcription factor complexes

Vivien I Strotmann [ID][1,6], Monica L García-Gómez[2,3,4] & Yvonne Stahl [ID][1,5,6✉]

## Abstract

**In *Arabidopsis thaliana* the root stem cell niche (SCN) is maintained by a complex regulatory network crucial for growth and developmental plasticity. However, many aspects of this network, particularly concerning stem cell quiescence and replenishment, remain unclear. Here, we investigate the interactions of key transcription factors (TFs) BRASSINOSTEROID AT VASCULAR AND ORGANIZING CENTRE (BRAVO), PLETHORA 3 (PLT3), and WUSCHEL-RELATED HOMEOBOX 5 (WOX5) in SCN maintenance. Analysis of mutants reveals their combinatorial regulation of cell fates and divisions in the SCN. In addition, studies using Fluorescence Resonance Energy Transfer Fluorescence Lifetime Imaging Microscopy (FRET-FLIM) in combination with novel analysis methods enable us to quantify protein–protein interaction (PPI) affinities and higher-order complex formation among these TFs. Our findings were integrated into a computational model, indicating that cell-type specific protein complex profiles and formations, influenced by prion-like domains in PLT3, play an important role in regulating the SCN. We propose that these unique protein complex signatures may serve as indicators of cell specificity, enriching the regulatory network that governs stem cell maintenance and replenishment in the *Arabidopsis* root.**

**Keywords** FRET-FLIM; Mathematical Modeling; Root Stem Cell Niche; Transcription Factor Complexes
**Subject Categories** Chromatin, Transcription & Genomics; Plant Biology; Stem Cells & Regenerative Medicine

## Introduction

As sessile organisms, plants must cope with environmental challenges and adapt their growth and development accordingly, as they cannot escape adverse conditions. The root system of higher plants plays a pivotal role for the plant's fitness, providing anchorage to the soil and access to water and nutrients. To ensure high developmental plasticity, plants maintain a reservoir of stem cells that reside in the root apical meristem (RAM) at the tip of the root. In *Arabidopsis thaliana* (*A. thaliana*), the center of the RAM harbors a group of slowly dividing, pluripotent stem cells termed the quiescent centre (QC). The QC exerts two key functions: first, it produces the surrounding tissue-specific stem cells (Heyman et al, 2014), also referred to as initials, through formative, asymmetric cell divisions, which occur either anticlinally or periclinally. These stem cells divide asymmetrically to give rise to different cell types from the outer to the inner layer: epidermis/lateral root cap, cortex, endodermis, pericycle and stele, as well as the columella at the root tip (Fig. 1G). The QC primarily divides periclinally thereby predominantly producing columella stem cells (CSC) (Cruz-Ramírez et al, 2013). Second, the QC acts as a signaling hub to maintain the surrounding stem cells in a non-cell autonomous manner (Benfey and Scheres, 2000; Dolan et al, 1993; van den Berg et al, 1997). The balance between QC quiescence and stem cell replenishment has to be maintained throughout the entire life cycle of a plant. This requires fine-tuned regulation, necessitating phytohormones, receptors and their ligands as well as several key transcription factors (TFs) (Strotmann and Stahl, 2021; García-Gómez et al, 2021).

The homeodomain TF WUSCHEL-RELATED HOMEOBOX 5 (WOX5) has been identified as a key regulator for stem cell maintenance in the root (Sarkar et al, 2007). By repressing *CYCLIN D3;3* (*CYCD3;3*) and *CYCLIN D1;1* (*CYCD1;1*), WOX5 inhibits cell divisions in the QC (Forzani et al, 2014). Furthermore, WOX5 maintains the undifferentiated status of the columella stem cells (CSCs) by repressing *CYCLING DOF FACTOR 4* (*CDF4*), which involves the recruitment of TOPLESS (TPL) and HISTONE DEACETYLASE 19 (HDA19) (Pi et al, 2015). Interestingly, WOX5 levels in the QC are fine-tuned through destabilization, such that higher WOX5 levels increase the number of columella cell layers (Cui et al, 2024). Recent findings suggest that to control the balance between maintaining the stem cell fate of CSCs and their differentiation, WOX5 also interacts with the auxin-dependent APETALA2-type TF PLETHORA 3 (PLT3) (Burkart et al, 2022). The *PLT* gene family comprises six members characterized as master regulators of root development (Aida et al, 2004; Galinha et al, 2007; Mähönen et al, 2014). While PLT5 and 7 are primarily involved in lateral root development (Hofhuis et al, 2013; Du and Scheres, 2017), PLT1-4 are expressed in the main root forming instructive protein gradients that are necessary for correct QC

[1]Institute for Developmental Genetics, Heinrich-Heine University, Universitätsstraße 1, 40225 Düsseldorf, Germany. [2]Theoretical Biology and Bioinformatics (IBB), Utrecht University, Padualaan 8, 3584 CS Utrecht, The Netherlands. [3]Experimental and Computational Plant Development (IEB), Utrecht University, Padualaan 8, 3584 CS Utrecht, The Netherlands. [4]CropXR Institute, Utrecht, The Netherlands. [5]Cluster of Excellence on Plant Sciences (CEPLAS), Heinrich-Heine University, Universitätsstraße 1, 40225 Düsseldorf, Germany. [6]Present address: Institute for Molecular Biosciences, Goethe University, Max-von-Laue Str. 9, 60438 Frankfurt am Main, Germany. ✉E-mail: y.stahl@bio.uni-frankfurt.de

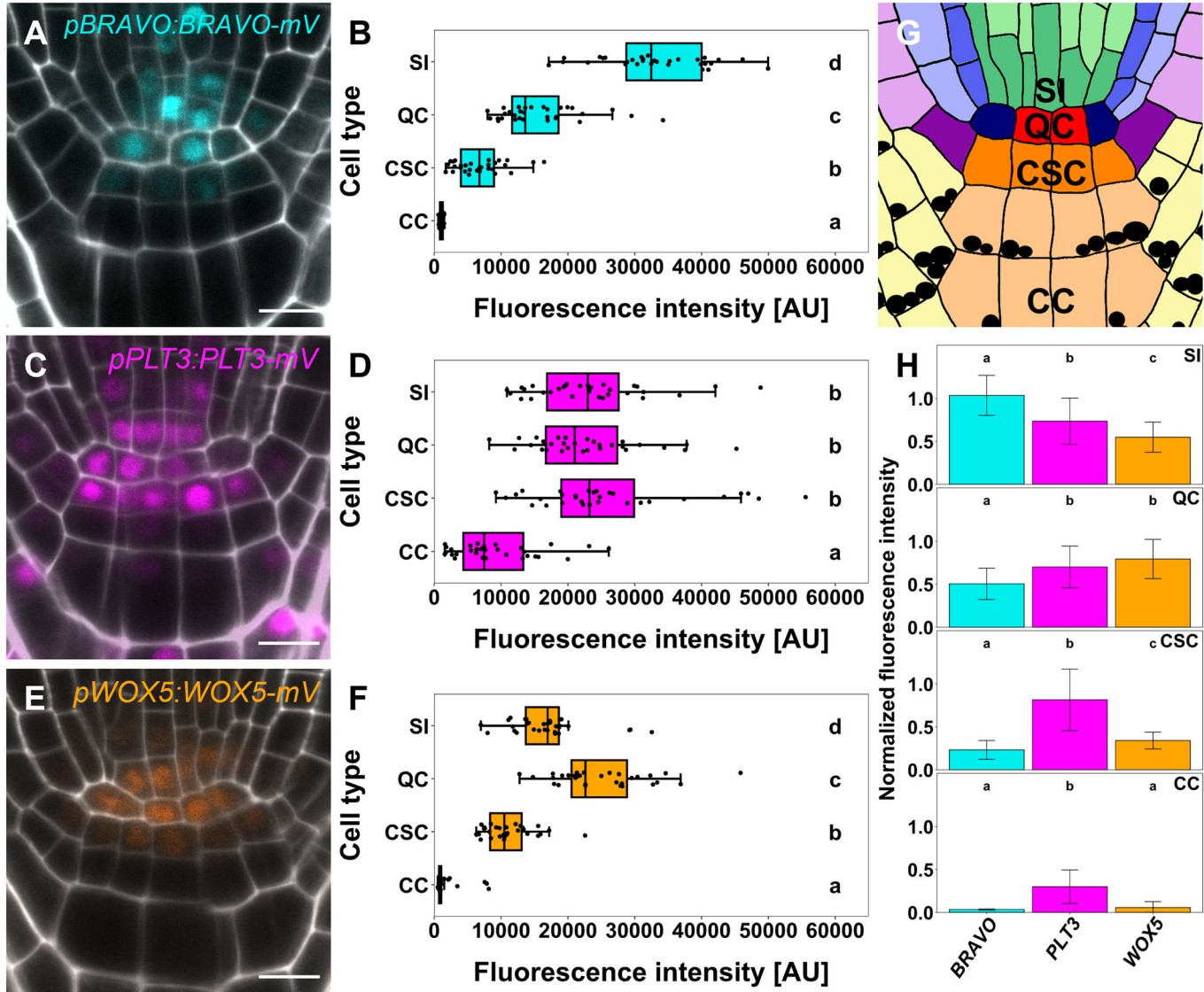

**Figure 1. Abundance of BRAVO, PLT3, and WOX5 in the Arabidopsis RAM.**

(A, B) Representative image of translational BRAVO reporter (A) and cell-type specific quantification of mVenus (mV) fluorescence intensity (B) in wild-type *Col-0* background in the RAM of Arabidopsis 5 DAG. (C, D) Representative image of translational PLT3 reporter (C) and cell-type specific quantification of mV fluorescence intensity (D) in wild-type *Col-0* background in the RAM of Arabidopsis 5 DAG. (E, F) Representative image of translational WOX5 reporter (E) and cell-type specific quantification of mV fluorescence intensity (F) in wild-type *Col-0* background in the RAM of Arabidopsis 5 DAG. (G) Schematic overview of the organization of the Arabidopsis RAM. The different cell types are represented by different colors. QC: red, cortex endodermis initial: dark blue, endodermis: mid blue, cortex: light blue, stele initials (SI): green, stele: light green, lateral root cap/epidermis initial: purple, epidermis: light purple, lateral root cap: light yellow, columella stem cell (CSC): orange and columella cell (CC): light orange. Starch granules are visualized as black dots. (H) Bar plot representing the mean fluorescence intensities of mV in BRAVO, PLT3, or WOX5 translational reporters in SIs, QCs, CSCs, and CCs normalized to the maximum intensity found for BRAVO in SIs. Error bars display standard deviation. Cell walls were stained using PI and are shown in white; expression of TF is visualized by mVenus in cyan (BRAVO), pink (PLT3), or orange (WOX5). Data information: DAG days after germination. (A, C, E) Scale bars: 10 μm. (B, D, F, H) Statistical groups were assigned after separate nonparametric Kruskal–Wallis with post hoc Dunn's test (α = 0.05, P values were adjusted after Benjamini and Hochberg). (B, D, F, H) For each translational reporter n = 28 (biological replicates) resulting from three technical replicates. (B, D, F) Box = middle 50% of data (= interquartile range (IQR)); whiskers = from IQR to min/max values, but at most 1.5 * IQR; line within box = median; data beyond the end of whiskers are "outliers" and are plotted independently. In addition, a jitter plot was used to visualize all data points individually.

positioning and cell fate decisions (Galinha et al, 2007; Aida et al, 2004; Mähönen et al, 2014). Interestingly, the loss of PLT3 or WOX5 function, as observed in *plt3-1* and *wox5-1* mutants, causes an increase in QC divisions (Sarkar et al, 2007; Pi et al, 2015; Burkart et al, 2022). This phenotype is even more severe in the *plt3 wox5* double mutant indicating that PLT3 and WOX5 act in

parallel pathways to control stem cell maintenance in the root (Burkart et al, 2022).

In the past decade brassinosteroids (BRs), a class of phytohormones, have been described to play an important role in the regulation of the root stem cell niche (SCN) maintenance (González-García et al, 2011). In the *Arabidopsis* RAM, BRs act through the R2R3-MYB TF

BRASSINOSTEROIDS AT VASCULAR AND ORGANIZING CENTRE (BRAVO) which inhibits QC divisions and is negatively regulated by the BR-dependent repressor complex formed by BRI1-EMS-SUPPRESSOR 1 (BES1) and TPL on transcript and protein level (Vilarrasa-Blasi et al, 2014; Espinosa-Ruiz et al, 2017). Recently, the ability of BRAVO to control formative QC divisions has been linked to WOX5 (Betegón-Putze et al, 2021), as *bravo-2* mutants, like *wox5-1* mutants, exhibit an increased frequency of QC divisions (Sarkar et al, 2007; Pi et al, 2015; Betegón-Putze et al, 2021; Burkart et al, 2022).

In addition to the described genetic interactions, one-on-one protein–protein interactions (PPIs) have been reported for WOX5 and PLT3 as well as for BRAVO and WOX5 (Betegón-Putze et al, 2021; Burkart et al, 2022). However, it is still unknown whether these TFs can also form higher-order complexes. In addition, it remains elusive how these genetic and physical interactions could potentially influence the regulation of stem cell maintenance. To unravel the underlying interplay of key TFs in the root SCN, we employed an integrative experimental and computational approach to analyze the protein complex formation between WOX5, PLT3, and BRAVO in the cells of the root SCN. Here, we demonstrate that cell-type specific profiles of protein complexes are formed and align their occurrence with the phenotypical SCN defects of the respective mutants. Moreover, by deleting specific interaction sites, we could demonstrate that heterodimerization contributes to maintaining stem cells in the root. Altogether, our results suggest that these unique protein complex 'signatures' convey cell-type specificity and could explain the different roles played by BRAVO, PLT3, and WOX5 in root SCN maintenance.

# Results

## BRAVO, PLT3, and WOX5 exhibit cell-type specific differences in protein abundance in the root SCN

First, we analyzed the absolute and relative abundance of BRAVO, PLT3, and WOX5 protein levels in the different cell types located in the SCN of the *Arabidopsis* root, focusing on the stele initials (SIs), QC, CSCs and columella cells (CCs) (Fig. 1G). This was achieved by measuring the fluorescence intensity of mVenus (mV) in the nuclei of the previously described *pPLT3:PLT3-mV* and *pWOX5:WOX5-mV* homozygous translational reporters in a *Col-0* WT background. Both reporters have been shown to complement their respective mutant phenotype (Burkart et al, 2022). In addition, we generated a homozygous, stable transgenic *Arabidopsis* line expressing *pBRAVO:BRAVO-mV* in the *Col-0* WT background, which is capable of complementing the *bravo-2* mutant phenotype (Appendix Fig. S1; Appendix Tables S7 and S8). We used the same microscopy settings for these quantifications to ensure that the detected protein levels are comparable. In line with previous findings, BRAVO protein levels are highest in the SIs and decrease significantly towards the CCs (Fig. 1A,B; Appendix Table S6) (Vilarrasa-Blasi et al, 2014). PLT3 levels are similar in SIs, QC, and CSCs, but significantly lower in the CCs (Fig. 1C,D; Appendix Table S6). WOX5 protein levels peak in the QC, decrease significantly in the adjacent SIs and CSCs, and are nearly absent in CCs (Fig. 1E,F; Appendix Table S6).

We summarized our findings in a protein abundance profile for each individual cell-type displaying the relative protein levels of BRAVO, PLT3, and WOX5. The protein levels are normalized to the overall maximum intensity observed for BRAVO in SIs (Fig. 1H). Accordingly, we found that BRAVO is the most abundant protein in SIs, followed by PLT3 and WOX5 in descending order. Conversely, in the QC, we observe a contrasting pattern, with WOX5 as the most abundant protein, followed by PLT3 and a significant reduction of BRAVO. PLT3 emerges as the predominant protein in the adjacent CSCs, along with significantly reduced levels of WOX5 and BRAVO protein. In differentiated CCs, WOX5 and BRAVO are almost absent and only low levels of PLT3 can be detected. Interestingly, although all of these regulators are expressed in several root SCN cells, our observations reveal quantitative differences in protein abundance that can be integrated into a cell-type specific "fingerprint". This provides a comprehensive snapshot of the unique protein levels within each cellular context, which could serve as an instructive output of cell-type specification (Fig. 1H).

## BRAVO, PLT3, and WOX5 jointly control CSC fate and QC divisions

Several studies have highlighted the inhibitory effect of BRAVO, PLT3 and WOX5 on QC divisions and CSC differentiation in the *Arabidopsis* root (Aida et al, 2004; Forzani et al, 2014; Galinha et al, 2007; Mähönen et al, 2014; Pi et al, 2015; Vilarrasa-Blasi et al, 2014). While all three proteins have been shown to be present in the QC and CSCs, a combinatory effect on QC division and CSC fate has only been investigated for WOX5 and PLT3 (Burkart et al, 2022) as well as for WOX5 and BRAVO (Betegón-Putze et al, 2021). Notably, such interplay not yet been observed for BRAVO and PLT3, nor for the simultaneous involvement of all three proteins.

To address this, we performed SCN stainings, that combine 5-ethynyl-2'-deoxyuridine (EdU) and modified pseudo Schiff base propidium iodide (mPSPI) stainings (Burkart et al, 2022), in several single and multiple mutants. This allowed us to analyze the differentiation status of the distal meristem, as well as the number of QC divisions that occurred within the last 24 h within the same root. To quantify CSC layers, the number of cell layers devoid of starch granules distally to the QC was counted. In *Col-0* WT, 68% of the roots show one CSC layer, whereas only 2% lack the starch-free CSC layer, and 30% show two CSC layers, most likely because they have recently divided (Figs. 2A,B,J and EV1A; Appendix Table S9). In *bravo-2* and *plt3-1* single mutants, the proportion of roots lacking the CSC layer increases to 11% and 12%, respectively, which is not significantly different from the WT as described previously (Fig. 2C,D,J and EV1B,C; Appendix Table S9) (Burkart et al, 2022; Betegón-Putze et al, 2021; Vilarrasa-Blasi et al, 2014). However, the proportion of roots showing no starch-free CSC layer significantly increased to 37% in *bravo plt3* double mutants (Figs. 2F,J and EV1F; Appendix Table S9). This additive effect suggests that PLT3 and BRAVO act in parallel pathways to control CSC differentiation. In 53% of the *wox5-1* mutant roots, the starch-free CSC layer is absent (Figs. 2E,J and EV1D; Appendix Table S9), highlighting the significance of WOX5 for CSC fate (Pi et al, 2015; Sarkar et al, 2007; Burkart et al, 2022). In addition, the *bravo wox5* and the *plt3 wox5* double mutants, as previously demonstrated (Betegón-Putze et al, 2021; Burkart et al, 2022), exhibit a significantly higher percentage of roots lacking the starch-free CSC layer, 90% and 74%, respectively, compared to the single mutants and the *bravo plt3* double mutants (Figs. 2G,H,J and EV1E,G; Appendix Table S9). On the other hand, the *bravo plt3 wox5* triple mutant, with

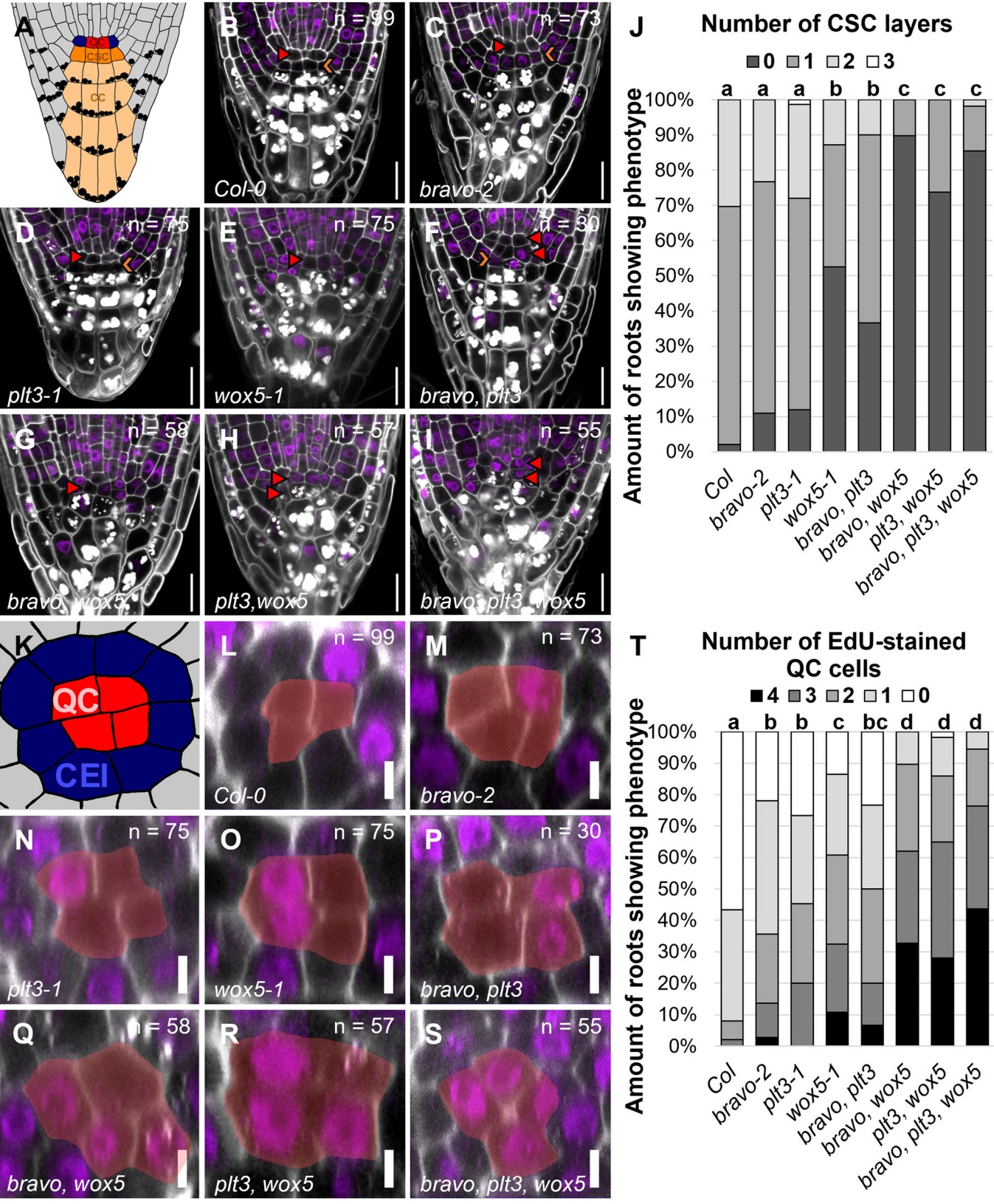

**Figure 2. BRAVO, PLT3 and WOX5 jointly regulate CSC differentiation and QC quiescence.**

(A) Schematic representation of a longitudinal section of the Arabidopsis RAM. Red: QC, blue: CEI, dark orange: CSC, light orange: CC. (B–I) Representative images of the mutant CSC phenotype in the indicated mutant background after combined mPSPI (white) EdU (purple) staining. The position of the QC is indicated by a red arrowhead and the CSC layer is marked with an orange arrowhead. Scale bars: 20 μm. (J) Quantification of SCN staining displaying 0, 1, 2, or 3 layers of CSC. (K) Schematic representation of a transversal section of the Arabidopsis RAM. QC cells are highlighted in red, and CEIs are displayed in blue. (L–S) Representative images of optical cross-sections of the Arabidopsis RAM in the indicated mutant background. The combined mPSPI/EdU staining reveals the cells that have divided within 24 h. QC is highlighted in yellow. Scale bars: 5 μm. (T) Quantification of SCN staining displaying 0, 1, 2, 3, or 4 or more QC divisions. Data Information: Arabidopsis seedlings were transferred to GM plates containing EdU 5 DAG and fixed after 24 h. (B–I, L–S) The numbers of analyzed roots for each genotype (biological replicates) are indicated in the respective microscopy images and results from 3 to 5 technical replicates. DAG days after germination. (J, T) Statistical groups were assigned after a nonparametric Kruskal–Wallis with post hoc Dunn's test ($\alpha = 0.05$, P values were adjusted after Benjamini and Hochberg).

85% of the roots lacking the CSC layer, resembles the *bravo wox5* and *plt3 wox5* double mutants (Figs. 2I,J and EV1H; Appendix Table S9). These results suggest that BRAVO, PLT3, and WOX5 jointly control CSC fate.

In addition, the quantification of QC divisions was performed by counting the number of EdU-stained nuclei within an optical transverse section through the RAM as described in (Burkart et al, 2022). QC cells were identified by their relative position within the RAM, situated directly below the vascular initials and surrounded by CEIs in a circular arrangement (Fig. 2K). In the WT, 57% of the roots do not show any QC cell divisions, while 35% show one QC cell division (Figs. 2L,T and EV1A; Appendix Table S9). In 6% and 2% of the analyzed roots, two and three QC divisions were observed, respectively, resulting in a total of 43% of the analyzed WT roots showing EdU-stained QC cells. In *bravo-2* and *plt3-1* single mutants, the proportion of roots showing at least one EdU-stained QC cell significantly increases to 78% and 73%, respectively, which is comparable to previous observations (Figs. 2M,N,T and EV1B,C; Appendix Table S9) (Betegón-Putze et al, 2021; Burkart et al, 2022). This phenotype is even more severe in *wox5-1* mutants, where at least one EdU-stained QC cell could be observed in 86% of the roots (Figs. 2O,T and EV1D; Appendix Table S9) (Burkart et al, 2022; Forzani et al, 2014). Like the above-described additive effects of CSC differentiation in the double and triple mutants, the number of roots showing at least one QC cell division increases to 100% and 98% in the *bravo wox5* and the *plt3 wox5* double mutants, respectively (Figs. 2P–R,T and EV1E–G, Appendix Table S9). In addition, the double mutants exhibit a significantly higher frequency of four divided QC cells in comparison to the respective single mutants: 7% in the *bravo plt3* double mutant, 36% in the *bravo wox5* double mutant and 30% in the *plt3 wox5* double mutant in comparison to 3%, 0% and 11% in the *bravo-2*, *plt3-1*, and *wox5-1* single mutants, respectively. A further yet not statistically significant increase in EdU-stained QC cells can be observed in the *bravo plt3 wox5* triple mutant where 44% of the roots display a completely divided QC (Figs. 2S,T and EV1H; Appendix Table S9). These observations indicate that BRAVO, PLT3 and WOX5 jointly control QC divisions, which may also involve other factors, e. g. SHORT-ROOT (SHR) and SCARECROW (SCR) (Cruz-Ramírez et al, 2013; Clark et al, 2020; Long et al, 2017).

Furthermore, we also examined if the QC exhibits additional periclinal cell divisions, which in *Col-0* WT occurs only in 4% of the roots (Fig. EV1I,K; Appendix Table S10). This phenotype manifests in 85% of *bravo-2* mutants (Fig. EV1J,K; Appendix Table S10). Additional periclinal cell divisions can also be observed in 43% of *plt3-1* single mutants and in 62% of *wox5-1* single mutants (Fig. EV1K; Appendix Table S10). In contrast to the number of EdU-stained QC cells, the frequency of periclinal cell divisions is relatively similar and not significantly different in the double or triple mutants, with 77%, 84%, 79%, and 85% of the roots showing additional periclinal cell divisions of the QC cells in the *bravo plt3*, *bravo wox5*, *plt3 wox5*, and *bravo plt3 wox5* mutants, respectively (Fig. EV1K; Appendix Table S10). This effect has already been described for *wox5-1* and *bravo-2* single mutants in comparison to the *bravo wox5* double mutant in earlier studies (Betegón-Putze et al, 2021).

To further investigate the consequences of SCN defects at 5 DAG (days after germination) on overall root development, we analyzed the total primary root length of these mutants at 10 DAG (Appendix Fig. S2; Appendix Table S11). Here we found that, in comparison to the *bravo plt3* double mutants the *bravo plt3 wox5* triple mutant roots are significantly shorter, once more highlighting the importance and combinatory effect of BRAVO, PLT3 and especially WOX5 for proper root meristem maintenance.

Taken together, our findings suggest a combinatory effect of BRAVO, PLT3, and WOX5 on QC division frequency and CSC fate decision affecting overall root growth.

## BRAVO, PLT3, and WOX5 can form a trimeric complex

In addition to the observed overlapping yet cell-type specific protein levels and the genetic interplay of BRAVO, PLT3 and WOX5, recent reports also provide evidence for one-on-one PPIs of BRAVO and WOX5, as well as for PLT3 and WOX5 (Burkart et al, 2022; Betegón-Putze et al, 2021). These findings raised the question of whether BRAVO and PLT3 could also interact. To address this, we first performed Fluorescence resonance energy transfer fluorescence lifetime imaging microscopy (FRET-FLIM) measurements in transiently expressing *Nicotiana benthamiana* (*N. benthamiana*) abaxial epidermal leaf cells using BRAVO-mV as donor molecule under the control of a β-estradiol inducible promoter as described earlier (Burkart et al, 2022). Results of FRET-FLIM measurements are often displayed as the average amplitude-weighted lifetime which is a mixture of differentially decaying components and is calculated by summing each component's lifetime weighted by its respective amplitude. In the case of FRET, the fluorescence lifetime decreases and serves as a measure for PPI. This reduction of lifetime results either from a large number of molecules that undergo FRET indicating a high affinity of the two proteins of interest (POIs) or a highly efficient energy transfer which demonstrates high proximity of the POIs and/or favorable fluorophore dipole orientation (Fig. 3A,B). The use of a novel analysis method allowed us to distinguish between these two scenarios, providing deeper insights into protein affinities, hereafter referred to as 'binding', between BRAVO, PLT3, and WOX5 (Maika et al, 2023; Orthaus et al, 2009).

The reference sample BRAVO-mV (donor-only control) shows an average binding of $2.3 \pm 7.4\%$ (Fig. 3C; Appendix Table S12) and the

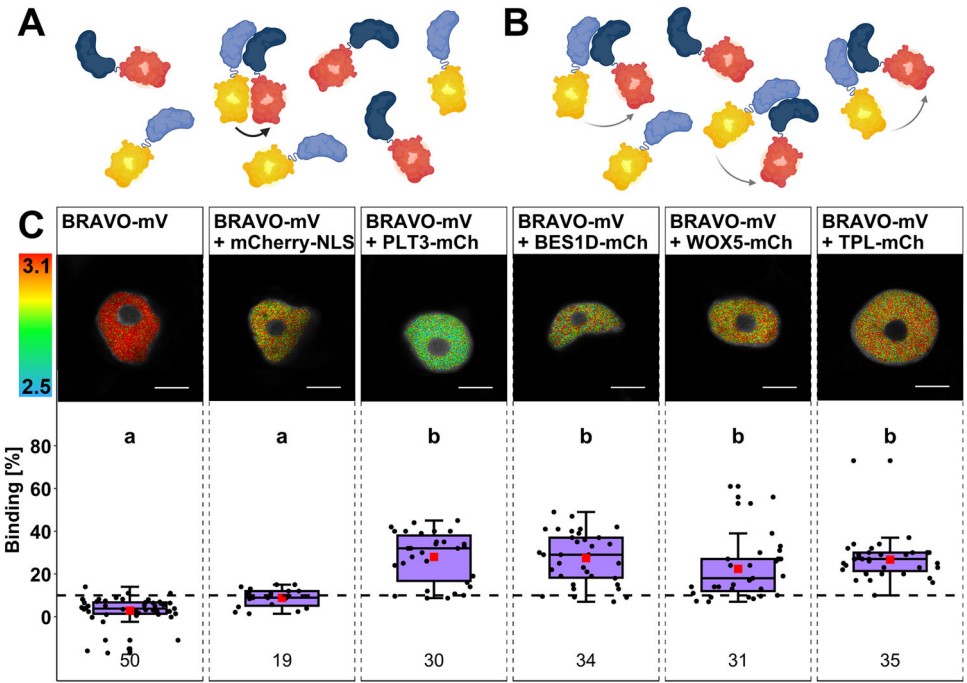

**Figure 3. BRAVO interacts with PLT3, WOX5, BES1D, and TPL.**

(A) A reduction of fluorescence lifetime as a consequence of FRET can be a result of a highly efficient energy transfer indicating close proximity. (B) On the other hand, a reduction of the fluorescence lifetime can result from a high affinity of the two proteins. (C) Upper panel: Representative images of fluorescence lifetime imaging microscopy (FLIM) measurements of nuclei in *N. benthamiana* epidermal leaf cells after pixel-wise multiexponential fitting. The fluorescence lifetime of the donor BRAVO-mV in the absence or presence of the indicated acceptor (of mCherry-NLS, PLT3-mCh, BES1D-mCh, WOX5-mCh, or TPL-mCh) is color-coded: blue (2.5) refers to low fluorescence lifetime [in ns], red (3.1) indicates high fluorescence lifetime [in ns]. Scale bars: 6 µm. Lower panel: Binding values [%] are represented as purple box plots of the same samples as in the upper panel. Statistical groups were assigned after a nonparametric Kruskal–Wallis with post hoc Dunn's test (α = 0.05, P values were adjusted after Benjamini and Hochberg). Mean values are visualized as red squares. The black dotted line indicates the Binding cutoff of 10%. The numbers of analyzed nuclei (biological replicates) are indicated below each sample and result from 3 to 5 technical replicates. Data information: (A, B) Created with BioRender.com and modified from (Maika et al, 2023). (C) Box = middle 50% of data ( = interquartile range (IQR)); whiskers = from IQR to min/max values, but at most 1.5 * IQR; line within box = median; data beyond the end of whiskers are 'outliers' and are plotted independently. In addition, a jitter plot was used to visualize all data points individually.

negative control composed of BRAVO-mV co-expressed with mCherry(mCh)-NLS shows a binding of 8.8 ± 4.3% (Fig. 3C; Appendix Table S12). A binding that is not significantly different from the negative control is regarded as showing no interaction. Upon co-expression of BRAVO-mV with PLT3-mCh, the binding significantly increases to 28.0 ± 11.7% (Fig. 3C; Appendix Table S12). To compare this observation with already confirmed interactions of BRAVO with WOX5 (Betegón-Putze et al, 2021), as well as with BES1 or TPL (Vilarrasa-Blasi et al, 2014), we co-expressed BRAVO-mV with WOX5-mCh or TPL-mCh resulting in binding values of 22.4 ± 14.1% and 26.7 ± 9.8%, respectively (Fig. 3C; Appendix Table S12) (Betegón-Putze et al, 2021). Interaction of BRAVO with BES1 was tested by co-expression of BRAVO-mV with BES1D-mCh, which was shown to mimic the dephosphorylated and thereby active form of BES1 and yielded an average binding of 27.4 ± 11.9%. This suggests similar affinities of BRAVO towards PLT3, BES1, and TPL, but a lower affinity towards WOX5 (Fig. 3C; Appendix Table S12).

These findings, along with previously described interactions of WOX5 with PLT3, TPL or BES1, as well as BES1 and TPL, prompted us to investigate, whether these TFs can also form higher-order complexes (Burkart et al, 2022; Espinosa-Ruiz et al, 2017; Vilarrasa-Blasi et al, 2014; Betegón-Putze et al, 2021). To address this, we used a combination of bimolecular fluorescence complementation (BiFC) and FRET (Fig. 4A,B) (Maika et al, 2023; Kwaaitaal et al, 2010). Here, the donor fluorophore is divided into two fragments: the N-terminal part of mVenus (mV(N)) and the C-terminal part (mV(C)). The interaction of WOX5 and PLT3, which has been described earlier (Burkart et al, 2022), has been shown to have a high affinity (Appendix Fig. S3 and Appendix Table S15). This is why we decided to tag WOX5 and PLT3 with mV(N) and mV(C), respectively. In this scenario, the interaction of WOX5 and PLT3 leads to the reconstitution of mV and restores its fluorescence, enabling us to perform FRET-FLIM when co-expressing another acceptor-labeled protein. The donor-only reference sample WOX5-mV(N) PLT3-mV(C) yields an average binding of 1.6 ± 14.1%, and the negative control WOX5-mV(N) PLT3-mV(C) with mCherry-NLS shows an average binding of 2.9 ± 5.0% (Fig. 4C; Appendix Table S13). Upon co-expression of BES1D-mCh or TPL-mCh, the binding significantly increases to 18.7 ± 8.0% and 23.3 ± 8.3%, respectively (Fig. 4C; Appendix Table S13). Notably, in the presence of BRAVO-mCh, the average binding strongly increases to 36.3 ± 10.7% (Fig. 4C; Appendix Table S13). Thus, the heterodimer of WOX5 and PLT3 shows higher affinity to BRAVO, which could suggest an increased probability and stability of the trimeric complex composed of WOX5, PLT3, and BRAVO compared to WOX5, PLT3, and BES1D, or TPL.

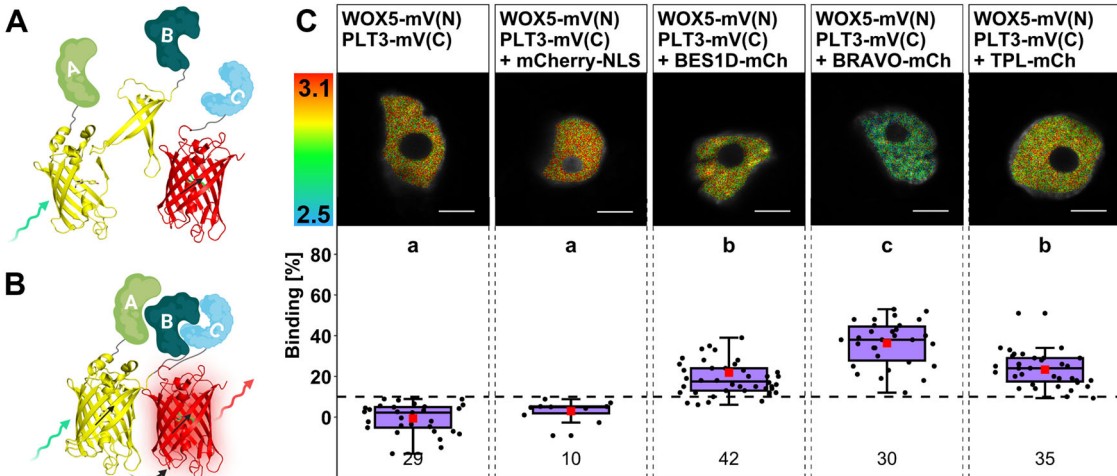

**Figure 4. Trimeric complex formation of WOX5 and PLT3 with BRAVO, BES1D, and TPL.**

(A) The combination of BiFC-FRET allows the detection of higher-order complexes. Here, the two fragments of a split donor fluorophore are fused to two proteins of interest (POI), while a third POI is fused to the acceptor. (B) In the case of trimeric complex formation, the donor molecule is reconstructed and transfer energy to the acceptor molecule by FRET after excitation. (C) Upper panel: Representative images of fluorescence lifetime imaging microscopy (FLIM) measurements of nuclei *N. benthamiana* epidermal leaf cells after pixel-wise multiexponential fitting. The fluorescence lifetime of the donor WOX5-mV(N)/PLT3-mV(C) in the absence or presence of the indicated acceptor (mCherry-NLS, BES1D-mCh, BRAVO-mCh or TPL-mCh) is color-coded: blue (2.5) refers to low fluorescence lifetime, red (3.1) indicates high fluorescence lifetime. Scale bars: 6 µm. Lower panel: Binding values [%] are represented as purple box plots of the same samples as in the upper panel. Statistical groups were assigned after nonparametric Kruskal–Wallis with post hoc Dunn's test (α = 0.05, P values were adjusted after Benjamini and Hochberg). Mean values are visualized as red squares. The black dotted line indicates the Binding cutoff of 10%. The numbers of analyzed nuclei (biological replicates) are indicated below each sample and result from 2 to 3 technical replicates. Data information: (A, B) Created with BioRender.com and modified from (Strotmann and Stahl, 2022). (C) Box = middle 50% of data ( = interquartile range (IQR)); whiskers = from IQR to min/max values, but at most 1.5 * IQR; line within box = median; data beyond the end of whiskers are "outliers" and are plotted independently. In addition, a jitter plot was used to visualize all data points individually.

To gain further insights into the potential of trimeric complex formation, we conducted additional FRET-FLIM measurements in *N. benthamiana* with rearranged fluorescent tags. Here, the donor fluorophore is shared between BRAVO and PLT3, namely BRAVO-mV(N) and PLT3-mV(C), which also showed high affinity (Fig. 3C; Appendix Table S12). The donor-only reference sample BRAVO-mV(N) PLT3-mV(C) exhibits an average binding of 2.2 ± 3.6% (Fig. EV2; Appendix Table S14). The negative control composed of BRAVO-mV(N) PLT3-mV(C) and mCherry-NLS shows a similar average binding of 3.2 ± 3.1% (Fig. EV2; Appendix Table S14). Surprisingly, co-expression of BES1D-mCh yields an average binding of only 10.2 ± 5.9% (Fig. EV2; Appendix Table S14), indicating that a trimeric complex composed of BRAVO, PLT3, and BES1D is unlikely to form. Contrary, the co-expression of TPL-mCh or WOX5-mCh leads to a significantly increased average binding of 21.1 ± 8.3% and 29.8 ± 10.6%, respectively (Fig. EV2; Appendix Table S14). This again suggests that a trimeric complex formed by BRAVO, PLT3, and WOX5 is more stable and occurs with a higher probability. Taken together, these findings reveal the formation of several combinations of protein multimers with different probabilities of occurrence as judged by their binding capacities. Here, the complex composed of BRAVO, PLT3 and WOX5 seems to be the most frequent and stable.

## Modeling reveals cell-type specific TF complex compositions

Our results reveal distinct, cell-type specific patterns of protein abundance for BRAVO, PLT3, and WOX5 in the root SCN (Fig. 1) along with the formation of diverse heterodimers with varying binding affinities as well as higher-order complexes in *N. benthamiana* (Figs. 3, 4, and EV2; Appendix Fig. S3). The protein complexes formed in the cells of the root SCN ultimately result from the cell-type specific protein levels and the binding affinities between the proteins. This raises the question whether dimerization and complex formation within the context of the root apex also display cell-type specificity, and how this is influenced by the protein levels in each cell of the SCN (Fig. 1). For example, BRAVO protein levels in the QC are notably lower compared to PLT3 or WOX5 (Fig. 1H), however, its consequence on protein complex formation remains undetermined. While the FRET-FLIM approach could in theory be used to investigate the formation of dimer- and oligomerization in *Arabidopsis* root cells, previous attempts to assess the interaction of PLT3 and WOX5 under the control of their endogenous promoter in roots have been challenging due to limited protein abundance and, consequently, low photon counts (Burkart et al, 2022). This is a limitation challenging to overcome without altering the endogenous protein levels. Therefore, as an alternative to identify potential TF specificity and cell-type specific complexes in the root SCN, we use a two-step mathematical modeling approach that combines the endogenous protein abundances (Fig. 1) with the binding probabilities for one-on-one PPIs and trimeric protein complexes (Figs. 3, 4, and EV2; Appendix Fig. S3). To enhance clarity on the workflow of our mathematical modeling approach, we included a supplementary figure that summarizes the fundamental steps of this process (Fig. EV3).

First, we performed a parameter analysis to predict the relative association and dissociation rates to form the WOX5-PLT3, BRAVO-PLT3, BRAVO-WOX5 heterodimers, and the WOX5-

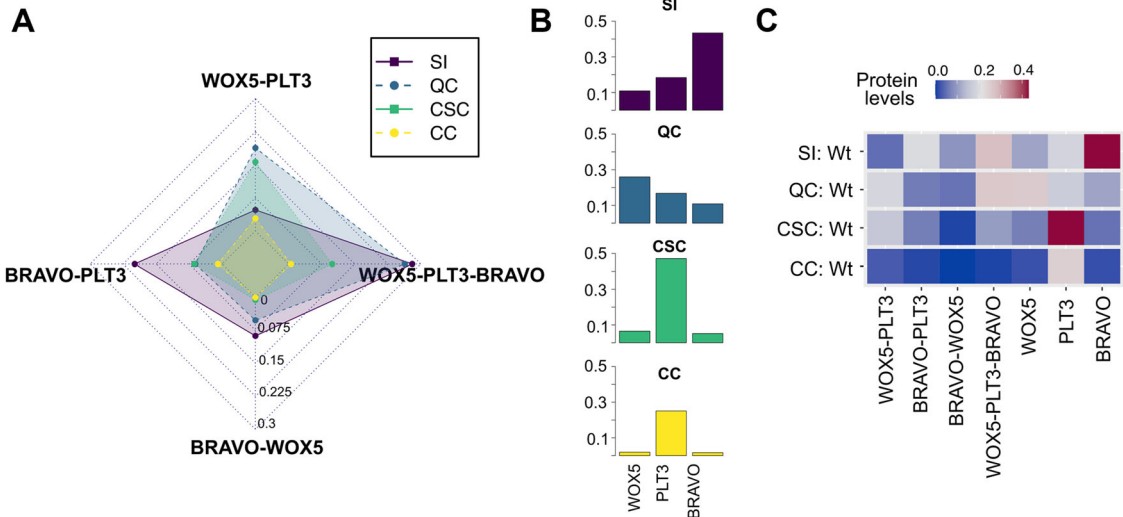

**Figure 5. In silico prediction of protein complex signatures in the WT root SCN.**

(A) Radar plot showing the levels of heterodimers and trimeric complex of WOX5, PLT3, and BRAVO formed in the SI (purple), QC (blue), CSC (green), and CC (yellow). The radial axis shows the protein levels (in arbitrary units). (B) Free WOX5, PLT3, and BRAVO protein in each of the simulated root SCN cells. (C) Heatmap showing the protein complexes and free protein in the cells of WT simulation. High concentrations are displayed in red, low concentration are displayed in blue. SI stele initals, QC quiescent center, CSC columella stem cells, CC columella cells.

PLT3-BRAVO trimeric complex. For the trimeric complex, we evaluate its formation via WOX5-PLT3 and BRAVO-PLT3 as donors (Figs. 4 and EV2). We start our simulations with equal levels of both donor and acceptor as initial condition, to mimic the conditions in the *N. benthamiana* experiments. Then, we simulate the protein complex formation using association and dissociation rates from a wide range of possible parameter values, until a steady state is reached. For each parameter combination tested, we evaluated if the proportion of protein in a steady-state complex corresponds to the value from the respective relative binding affinity determined with our experiments. Repeating this parameter estimation for each of the protein complexes under study allows us to identify several parameter combinations capable of producing protein complexes in line with FRET-FLIM experimental data (Appendix Figs. S4 and S5A). The predicted parameter combination for protein complexes with a high binding affinity (i.e., WOX5-PLT3) falls in the space where association rates are higher than the dissociation rates (Appendix Fig. S4), in contrast to lower binding affinity complexes (i.e., BRAVO-WOX5). These determined parameters allow us to describe our binding experimental data in a computational model.

Next, we simulated the protein complexes formed by BRAVO, WOX5, and PLT3 in each of the cells of the root SCN (Fig. EV3). For this, we use as initial condition the values from the relative fluorescence intensities reflecting the protein abundances of BRAVO, PLT3 and WOX5 in the SI, QC, CSC, and CC (Fig. 1H), and the association/dissociation rates per complex from our parameter analysis. Therefore, the cell-type specific profiles of protein complexes predicted by modeling are the emergent result of how much protein is available in each cell-type and the binding affinities between specific protein pairs and complexes (Fig. 5). We summarized these results in a radar chart where the level of each protein complex is arranged in a different radial axis and displayed

free protein levels that remain after complex formation are displayed separately as bar plots (Fig. 5A,B). Furthermore, we combined these results in a heatmap (Fig. 5C). In addition, we performed several controls that assume different combinations of experimental data, both binding affinities and cell-type specific protein abundances, and varying ratios of association and dissociation (Appendix Fig. S6, "Methods"). Interestingly, results comparable to our model were only observed in control 2, assuming higher association and dissociation rates, which also indicates higher association in our experimental data.

Our simulation reveals that SIs and QC cells are characterized by high levels of the trimeric protein complex WOX5-PLT3-BRAVO, which reflects their combinatory role in the regulation of QC divisions (Fig. 5A,C). The CSC cells are predicted to be enriched in the WOX5-PLT3 complexes. Such enrichment could be related to the previously described function of the WOX5-PLT3 complex in CSC maintenance (Burkart et al, 2022). Finally, the CCs are predicted to have negligible levels of all protein complexes studied, consistent with the very low BRAVO, PLT3, and WOX5 protein levels in these cells according to our quantification (Fig. 1). Notably, these protein complex "signatures" are strikingly different in each of the simulated cells and the resulting polygons are unique for each cell type (Fig. 5A), which might be related to their specific function.

Curiously, the levels of free protein also show cell-type specific patterns that allow us to further distinguish between SIs, QC, and CSCs (Fig. 5B,C). SIs are enriched in free BRAVO, while the QC shows high levels of free WOX5. Both, CSCs, and CCs, exhibit high levels of PLT3. It is interesting to consider that these free proteins could participate in both, binding other proteins not considered here, and/or intercellular movement, assuming an increased mobility if the protein is not in complexes (Fig. 5B,C). For instance, the levels of free WOX5 in the QC cells could constitute a

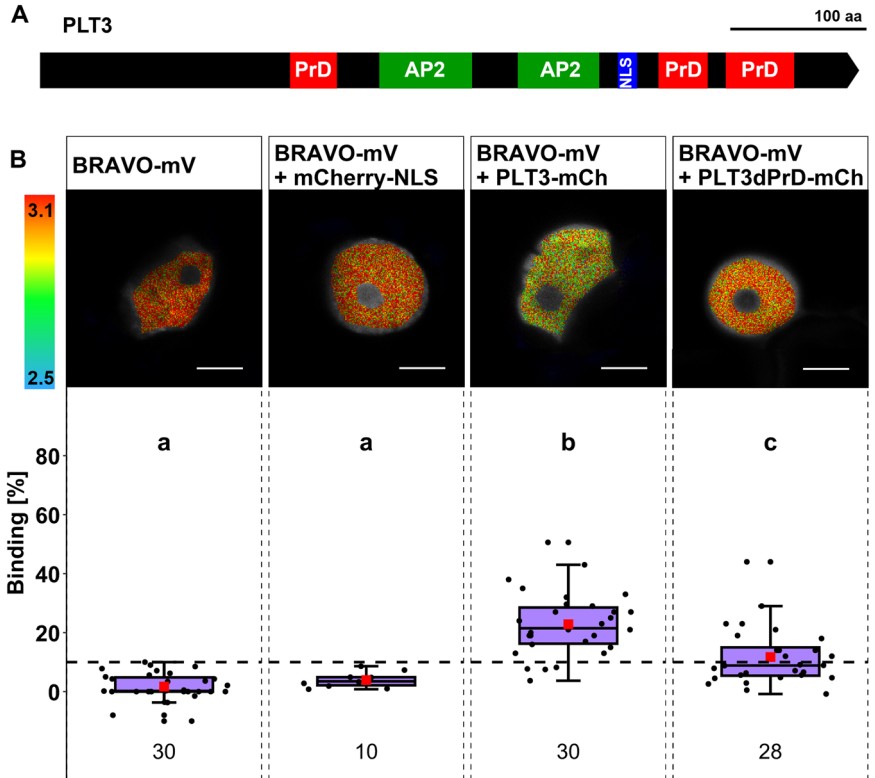

**Figure 6. PrDs of PLT3 stabilize interaction with BRAVO.**

(A) Schematic of PLT3 aa sequence. Red: PrDs; green: APETALA2(AP2)-domain; blue: nuclear localizing sequence (NLS). (B) Upper panel: Representative images of fluorescence lifetime imaging microscopy (FLIM) measurements of nuclei in *N. benthamiana* epidermal leaf cells after pixel-wise multiexponential fitting. The fluorescence lifetime of the donor BRAVO-mV in the absence or presence of the indicated acceptor (mCherry-NLS, PLT3-mCh, or PLT3dPrD-mCh) is color-coded: blue (2.5) refers to low fluorescence lifetime [in ns], red (3.1) indicates high fluorescence lifetime. Scale bars: 6 μm. Lower panel: Binding values [%] are displayed as purple box plots of the same samples as the upper panel. Statistical groups were assigned after nonparametric Kruskal–Wallis with post hoc Dunn's test (α = 0.05, *p* values adjusted after Benjamini and Hochberg). Mean values are visualized as red squares. The black dotted line indicates the Binding cutoff of 10%. The numbers of analyzed nuclei (biological replicates) are indicated below each sample and result from 2 to 3 technical replicates. Data Information: (B) Box = middle 50% of data ( = interquartile range (IQR)); whiskers = from IQR to min/max values, but at most 1.5 * IQR; line within box = median; data beyond the end of whiskers are "outliers" and are plotted independently. In addition, a jitter plot was used to visualize all data points individually.

pool of free protein available for intercellular mobility towards the neighboring CSCs as previously described (Pi et al, 2015; Berckmans et al, 2020). In summary, these results support the hypothesis that complex formation, especially heteromerization, occurs in a cell-type specific context.

## Prion-like domains of PLT3 serve as conserved interaction hub

After we have found evidence for the formation of TF complexes with cell-type-dependent variations, we asked whether these complexes are important for root SCN maintenance. To address this, we aimed to destabilize the interaction of these TFs by mutating their specific interaction sites and observing if this altered protein can still rescue the phenotypical defects in the SCN. First, we explored the literature to identify potential interaction sites of BRAVO, PLT3, and WOX5. Previous studies have shown that prion-like domains (PrDs) in PLT3 mediate the interaction with WOX5 (Fig. 6A) (Burkart et al, 2022). PrDs are intrinsically disordered regions (IDRs) and serve not only as mediators of multivalent interactions but have also been demonstrated to be

involved in chromatin opening (Levy et al, 2002) and phase separation (Jung et al, 2020; Fang et al, 2019; Wang et al, 2022). Given the presence of PrDs also in PLT1, PLT2, and PLT4, albeit in lower numbers (Burkart et al, 2022), we hypothesized that these regions function as conserved interaction sites. Thus, we deleted the two C-terminal PrDs and replaced the N-terminal PrD with an alanine-glycine linker, termed PLT3ΔPrD, and performed FRET-FLIM measurements to investigate how the loss of PLT3 PrDs affects its interaction with BRAVO. The donor-only reference control BRAVO-mV yields an average binding of 1.7 ± 4.6%, which increases not significantly to 3.9 ± 2.4% in the presence of mCherry-NLS serving as negative control (Fig. 6B; Appendix Table S16). Upon co-expression of BRAVO-mV with PLT3-mCh, the binding significantly increases to 22.8 ± 10.5% (Fig. 6B; Appendix Table S16). However, BRAVO-mV co-expressed with PLT3ΔPrD-mCh yields an average binding of only 11.7 ± 9.6%, suggesting that the deletion of the PrDs significantly reduces the interaction of PLT3 with BRAVO. Furthermore, we reanalyzed the published data of WOX5 interaction with PLT3 and PLT3ΔPrD (Burkart et al, 2022) revealing that interaction of WOX5 and PLT3 is not completely abolished upon loss of PrDs but significantly reduced

from 30.6 ± 13.3% with full-length PLT3 to 18.5 ± 12.4% in the absence of PLT3 PrDs (Appendix Fig. S3; Appendix Table S15).

To further support our hypothesis that the PrDs in PLTs act as conserved interaction sites, we investigated whether PLT3 also interacts with BES1 and TPL, which were shown before to interact with BRAVO and WOX5 (Vilarrasa-Blasi et al, 2014; Pi et al, 2015; Betegón-Putze et al, 2021) and if the deletion of PLT3 PrDs can also alter this interaction. To address this, we conducted FRET-FLIM in the presence of an acceptor-labeled PLT3 or PLT3ΔPrD. For the donor-only reference control measurements BES1D-mV, an average binding of 0.0 ± 6.3% could be observed, significantly increasing to 16.9 ± 8.1% in the presence of PLT3-mCh indicating PPI (Fig. EV4A; Appendix Table S17). However, co-expression of BES1D-mV with PLT3ΔPrD-mCh shows a reduced binding of 8.4 ± 4.8% which is not significantly different from the negative control BES1D-mV with mCherry-NLS exhibiting an average binding of 4.9 ± 6.5% (Fig. EV4A; Appendix Table S17). The reference control TPL-mV exhibits an average binding of 0.6 ± 5.5%, increasing to 6.4 ± 2.4% when co-expressed with the negative control mCherry-NLS (Fig. EV4B; Appendix Table S18). Upon co-expression of TPL-mV with PLT3-mCh, the average binding significantly increases to 13.5 ± 4.3%, suggesting a moderate interaction of TPL with PLT3 (Fig. EV4B; Appendix Table S18). Similar to BES1, the interaction of TPL and PLT3 is also abolished by the deletion of PrDs, demonstrated by a significantly decreased average binding of 8.99 ± 5.26% for TPL-mV with PLT3ΔPrD-mCh (Fig. EV4; Appendix Tables S17 and S18). In summary, these findings support the idea that the PrDs of PLT3 serve as a conserved interaction site for numerous TFs present in the root SCN.

## Redistribution of TF complexes alters the regulation of QC divisions

Next, we aimed to analyze the functional relevance of the eliminated or reduced interaction of PLT3 with other TFs present in the *Arabidopsis* root by deleting its PrDs. To address this, we created two transgenic *Arabidopsis* lines, using either full-length PLT3 or PLT3ΔPrD C-terminally tagged with mTurquoise2 (mT2) in combination with the dexamethasone (DEX) inducible ligand-binding domain of the rat glucocorticoid receptor (GR) in the *plt3-1* mutant background. Using the *WOX5* promoter allowed us to specifically investigate how the loss of PLT3 PrD influences QC maintenance. These lines were named *pWOX5:GR-PLT3-mT2* (*pWOX5:iPLT3*) and *pWOX5:GR-PLT3ΔPrD-mT2* (*pWOX5:iPLT3ΔPrD*). Finally, we performed a SCN staining and investigated if the QC exhibited additional periclinal cell divisions after inducing the plants by DEX treatment or in the presence of dimethyl sulfoxide (DMSO), which serves as a control (Fig. 7A–I; Appendix Table S19).

Under control conditions, only 27% of *Col-0* WT roots show additional periclinal cell divisions in the QC, which does not change significantly in the presence of DEX (Fig. 7A,E,I; Appendix Table S19). In agreement with previous observations (Burkart et al, 2022), *plt3-1* single mutant roots show additional periclinal cell divisions in the QC of 73% under control conditions and 87% after induction with DEX (Fig. 7B,F,I; Appendix Table S19). In *pWOX5:iPLT3* and *pWOX5:iPLT3ΔPrD* transgenic lines, 83% and 94% of the roots exhibit a periclinal cell division in the QC under control conditions, respectively, which is even higher than the *plt3-1* single mutant (Fig. 7C,D,I; Appendix Table S19). However, in the presence of DEX, the number of roots expressing *pWOX5:iPLT3*

showing this phenotype is reduced to 67%, which is, however, not significantly different from the DMSO-treated control, but still indicates that full-length PLT3 in the QC has the potential to partially restores the *plt3-1* periclinal cell division phenotype (Fig. 7G,I; Appendix Table S19). Contrary, the observed over-proliferated phenotype that we see under control conditions in *pWOX5:iPLT3ΔPrD* mutant roots is unaffected in the presence of DEX, indicating that the PrDs of PLT3 may be essential to inhibit additional periclinal QC divisions and thereby contribute to PLT3 function in root SCN maintenance (Fig. 7H,I; Appendix Table S19).

After observing the reduced affinity of PLT3ΔPrD for BRAVO and WOX5 and that it was unable to rescue SCN defects in *plt3-1* single mutants, we decided to use our computational model to predict immediate changes in the protein complex "signatures" in the root SCN that may have contributed to this failed rescue. Thus, we simulated the protein complex formation in the SI, QC, CSC, and CC as described before but adjusted the association rates for WOX5-PLT3ΔPrD according to the analysis of published data (Burkart et al, 2022) (Appendix Fig. S3; Appendix Table S15) and use the binding affinity we have determined experimentally for BRAVO-PLT3 ΔPrD (Fig. 6B; Appendix Table S16). This leads to a dramatic shift in the protein complex "signatures" of the root SCN cells (Fig. 7J–L). The reduced affinity of PLT3ΔPrD toward WOX5 and BRAVO alters dimer formation, causing a redistribution of BRAVO and WOX5 to the other protein complexes and an increase of free BRAVO, PLT3, and WOX5 protein levels in nearly all simulated cell types (Fig. 7J,K). While the BRAVO-PLT3 and WOX5-PLT3 complexes can still be formed, their levels are noticeably reduced in the SI, QC, and CSC cells. Furthermore, the BRAVO-WOX5 complex levels slightly increase in the SI and QC cells. Surprisingly, the abundance of the trimeric complex in SIs and the QC was only mildly affected. Here, in line with our modeling approach, we could experimentally verify by FRET-FLIM measurements of WOX5-mV(N) and BRAVO-mV(C) in the presence of either PLT3-mCh or PLT3ΔPrD-mCh, yielding an average binding of 31.5 ± 8.5% and 28.9 ± 8.4%, respectively (Appendix Fig. S7; Appendix Table S20). Altogether, our PLT3ΔPrD simulation provides insights into the alterations in cell-type specific protein levels that could be causative for defects observed experimentally in the root SCN.

# Discussion

In the past decades, our understanding of stem cell function and maintenance in the root of *Arabidopsis* has witnessed significant advances. Various aspects, including hormonal, developmental, as well as stress-related mechanisms have been discovered (García-Gómez et al, 2021; Strotmann and Stahl, 2021; Ubogoeva et al, 2021; Nolan et al, 2020). However, the underlying intricate network of molecular factors remains largely enigmatic. In this study, we aimed to unravel a new aspect of the regulatory network that controls root SCN maintenance related to protein complex formation.

Utilizing the SCN staining method (Burkart et al, 2022) allowed us to assess phenotypical SCN architecture defects of several single and multiple mutants (Fig. 2). Similar to previous findings, we observed a relatively moderate increase in the CSC differentiation and QC division phenotypes in *bravo-2* and *plt3-1* single mutants (Galinha et al, 2007; Burkart et al, 2022; Vilarrasa-Blasi et al, 2014; Betegón-

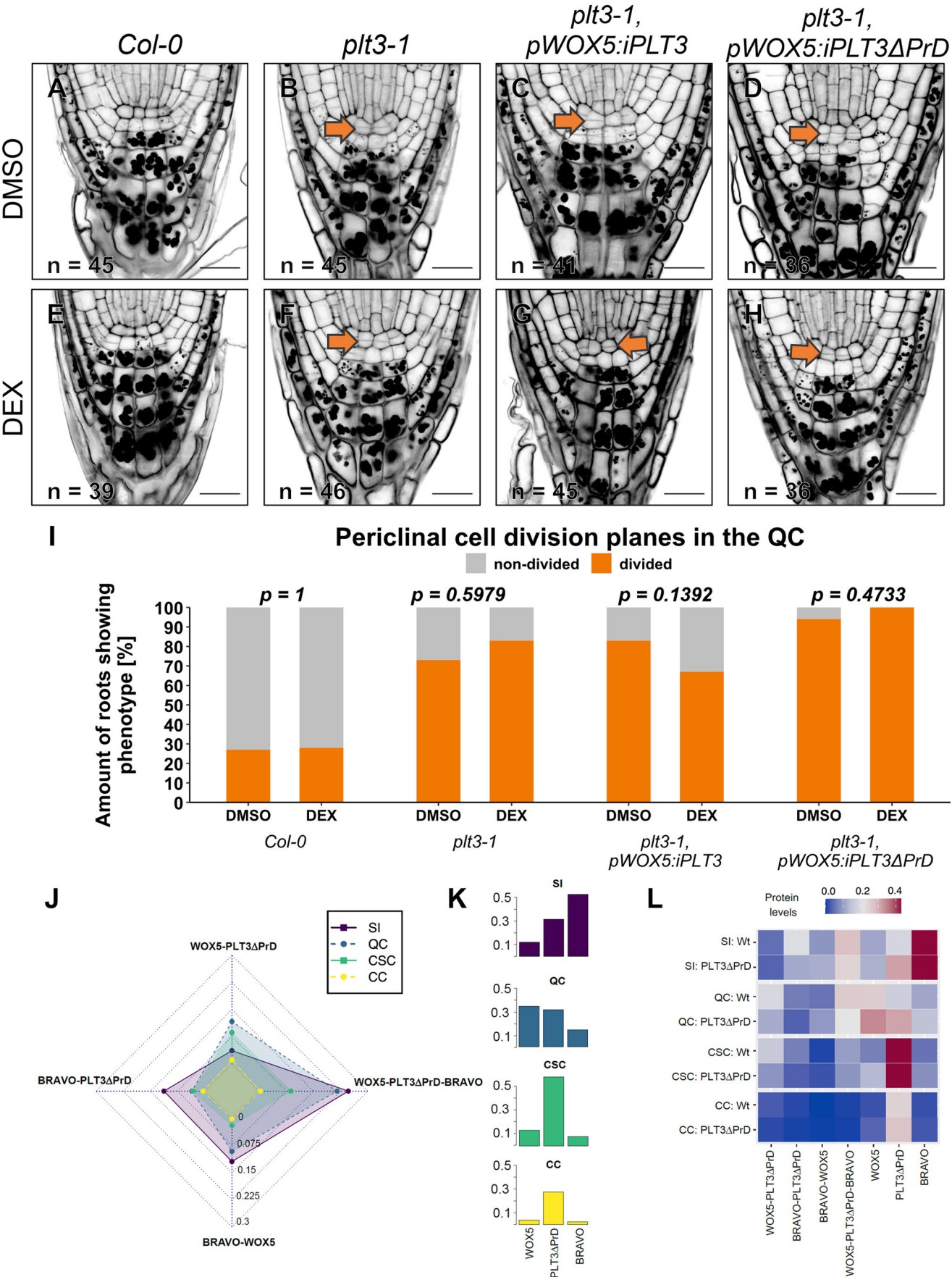

**I    Periclinal cell division planes in the QC**

◀ **Figure 7. PLT3 PrDs inhibit periclinal QC divisions and in silico predicted protein complex signatures in the root SCN.**

(A–H) Representative images of the *Arabidopsis* root meristem 6 DAG showing additional periclinal cell divisions in the QC in the absence (A–D) or presence (E–H) of DEX in the indicated genetic background. Divided QCs are highlighted with orange arrows. Scale bars: 20 μm. DAG days after germination. (I) Quantification of periclinal cell divisions when roots are treated with DMSO or DEX. Correlation between periclinal QC divisions and treatments was tested with Pearson's Chi-squared test with Yates' continuity correction. The number of analyzed roots (biological replicates) are indicated in each representative image and result from 3 technical replicates. (J) Radar plot showing the levels of heterodimers and trimeric complex between WOX5, PLT3ΔPrD, and BRAVO formed in the stele initials (purple), QC (blue), CSC (green), and CC (yellow). (K) Free WOX5, PLT3ΔPrD, and BRAVO protein in each of the simulated root SCN cells. (L) Heatmap showing the protein complexes and free protein in the cells of WT and PLT3ΔPrD simulations, the profiles are visibly different with a marked increase in free PLT3 in the CSC. High concentrations are displayed in red, low concentrations are displayed in blue. SI stele initials, QC quiescent center, CSC columella stem cells, CC columella cells.

Putze et al, 2021), and a more severely defective root SCN in *wox5-1* single mutants (Sarkar et al, 2007; Pi et al, 2015; Cruz-Ramírez et al, 2013; Betegón-Putze et al, 2021; Burkart et al, 2022; Berckmans et al, 2020). While in case of PLT3 such differences can be attributed to the substantial redundancy within the PLT TF family (Burkart et al, 2022; Galinha et al, 2007), for BRAVO and WOX5 the differences in intensity of the QC and CSC defects correlate with their relative protein levels in these cells (Fig. 1). Moreover, the stronger effects observed in the double and triple mutants containing *wox5-1* (i.e., *bravo wox5, plt3 wox5* and *bravo plt3 wox5*) highlight that WOX5 plays a critical role in the regulatory network underlying QC division and CSC differentiation (Fig. 2).

The genetic interplay of BRAVO, PLT3 and WOX5 regarding root SCN maintenance prompted us to evaluate their physical interaction (Fig. 3). While interactions of PLT3 and WOX5, as well as BRAVO and WOX5, have been described before (Betegón-Putze et al, 2021; Burkart et al, 2022), evidence for an interaction of PLT3 and BRAVO was still missing. Our results reveal for the first time PPI between BRAVO and PLT3 and between PLT3 and BES1 and TPL (Figs. 3 and EV4). Together with previously described, independent one-on-one interactions, these findings support the hypothesis of three pathways that control CSC differentiation and QC divisions in parallel.

The combination of BiFC and FRET allowed us to investigate the formation of higher-order complexes (Figs. 4 and EV2), revealing that of the trimeric complexes tested the complex WOX5-PLT3-BRAVO appears to be the most abundant. The observed variations of stability and probability of occurrence as indicated by a special analysis tool (Orthaus et al, 2009; Maika et al, 2023), could indicate a specific mechanism that facilitates the interaction of two or more POIs in a highly dynamic microenvironment where the number of proteins is generally high, such as in the QC (Fig. 1). The heterodimerization of transcriptional regulators increases binding specificity and affinity and allows the combination of different internal as well as external signal inputs into gene regulation (Strader et al, 2022). This idea is reinforced when considering that both the auxin-regulated WOX5 and BR-dependent BRAVO have been demonstrated to control the same cell cycle-related genes (*CYCD1;1, CYCD3;3*) (Vilarrasa-Blasi et al, 2014; Forzani et al, 2014). So far, cell cycle-related downstream targets of PLT3 remain unknown, yet, it is possible that PLT3 regulates these cyclins when associated with WOX5 or BRAVO. Further investigations are necessary to uncover potentially common downstream targets of BRAVO, PLT3 and WOX5.

Next, we used a computational modeling approach to elaborate on the consequences of the differences in protein abundance (Fig. 1H) and the binding affinities (Figs. 3, 4, 6B and EV2; Appendix Fig. S3) on the formation of protein complexes in cells of the root SCN. This

strategy allowed us to describe cell-type specific protein complex profiles in WT roots that might reflect how shared targets are regulated in each cell type (Fig. 5). Namely, high levels of the WOX5-BRAVO-PLT3 trimeric complex appear to be characteristic for stele initials and QC cells, which can be further discriminated by high free BRAVO protein abundance in the SIs and high free WOX5 levels in the QC. Several studies highlighted the elevated abundance of WOX5 in the QC, which could be either linked to interactions with other proteins not analyzed here or its non-cell autonomous function in the adjacent initials, although its necessity as a mobile stemness factor is still under debate (Pi et al, 2015; Berckmans et al, 2020). The CSCs were predicted to be enriched in the WOX5-PLT3 heterodimer, which aligns with their previously described impact on CSC differentiation (Burkart et al, 2022). Furthermore, CSCs possess high levels of PLT3 (Fig. 5), which might be linked to the nuclear body (NB) formation of PLT3, which was linked to its PrDs and is concentration dependent and may involve PLT3 homomerization. This mechanism could facilitate the recruitment of the WOX5-PLT3 heterodimer into these pre-formed NBs, as demonstrated previously (Burkart et al, 2022). Lastly, in CCs, the absence of BRAVO and WOX5 hinders complex formation, resulting in high levels of free PLT3 (Fig. 5). Compared to CSC, PLT3 levels are notably lower, accompanied by a loss of NB formation (Burkart et al, 2022). This implies that a specific protein concentration is required to trigger NB formation initially, highlighting the difference between differentiated CCs and the stem cell fate determination process in CSCs. Altogether, our findings imply the formation of dimers that together with differences of free protein levels convey cell-type specificity in the root (Fig. 8).

Interestingly, the additive effects observed in the *plt3 wox5* double mutants previously led to the hypothesis that WOX5 and PLT3 act in parallel pathways to maintain the integrity of the root SCN (Burkart et al, 2022). Our findings suggest that the additional pathways involve the combinations BRAVO and PLT3, as well as BRAVO and WOX5, along with the formation of a trimeric complex. This indicates that these TFs could act in four independent constellations to regulate SCN maintenance depending on which protein they are partnering with, which is determined by the protein levels in each cell of the SCN. Importantly, the role of the TFs studied here may partially contribute to other pathways that inhibit QC divisions supported by the observed additive phenotype of the triple mutant compared to the double mutants (Fig. 2). For instance, cytokinin induces the destabilization of WOX5, and higher levels of WOX5 in the QC result in an increase in the number of CSCs. It is tempting to speculate that such higher WOX5 levels could affect the protein complexes profile of the QC cells, resulting in a higher proportion of free WOX5 that, if transported to the adjacent cell layers, could produce multiple

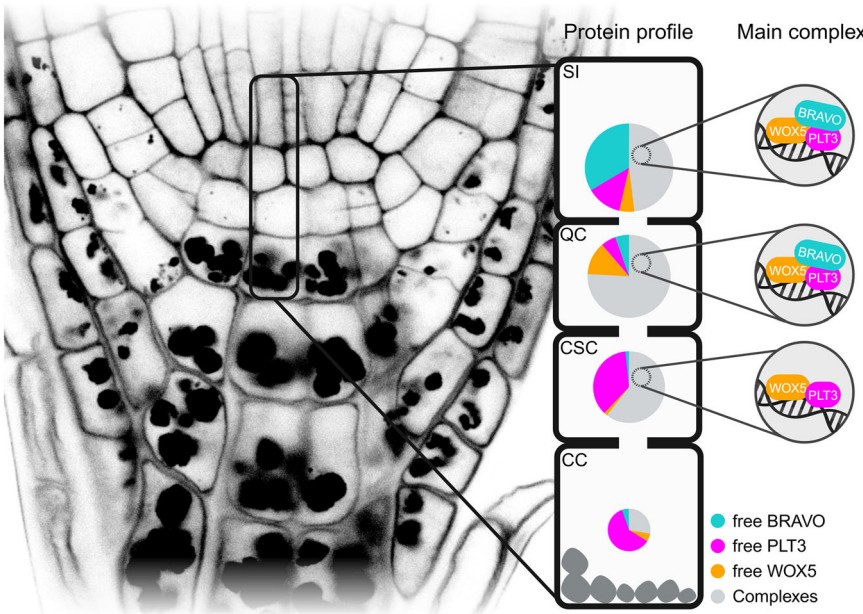

**Figure 8. Model of protein signatures and complexes in the root SCN.**

The nuclei of different cell types (SI, QC, CSC, CC) show distinct protein profiles of free BRAVO (turquoise), PLT3 (magenta), and WOX5 (orange) protein levels and main complexes (gray and insets). The size of the pie chart reflects the overall protein concentration in the nuclei of the specific cell type from high concentration (big) to low concentration (small). Created with BioRender.com.

layers of CSCs (Cui et al, 2024). Additional functions in other independent pathways have already been described for WOX5 in the SHR-SCR regulatory network (Clark et al, 2020; Cruz-Ramírez et al, 2013; Zhai et al, 2020). In addition, TEOSINTE-BRANCHED/ CYCLOIDEA/PCNA 20 (TCP20) was found to mediate the interaction of PLT3 and SCR, to specify the QC and establish the root SCN (Shimotohno et al, 2018). If and to what extent these molecular factors genetically and physically interact with other TFs in the SCN will be an interesting perspective for future investigations.

To investigate the impact of heterodimer- and oligomerization on root SCN maintenance, we aimed to identify potential interaction sites in the BRAVO, PLT3 or WOX5 amino acid sequence. Previous studies revealed that PrDs found in PLT3 act as mediators of its interaction with WOX5 (Burkart et al, 2022). PrDs are also present in PLT1,2 and 4, accompanied by NB formation. However, PLT3 harbors the highest number of PrDs, which correlates with stronger NB formation compared to PLT1, 2, and 4. Here, we demonstrated that loss of PrDs also negatively influences PLT3 interaction with WOX5, BRAVO, BES1, and TPL (Figs. 6 and EV4; Appendix Fig. S3). These findings suggest that PrDs act as a multivalent interaction hub, indicating a conserved function among other PLTs. Furthermore, our findings suggest that loss of PLT3 PrDs negatively affects its ability to inhibit periclinal QC divisions by demonstrating that PLT3ΔPrD, expressed in the QC, is unable to rescue the *plt3-1* periclinal QC division phenotype while full-length PLT3 partially reduced periclinal QC divisions (Fig. 7). Together, this could indicate that correct oligomerization is necessary for proper QC maintenance. We integrated our findings into our model by updating the binding affinities of PLT3ΔPrD

with BRAVO and WOX5. Here, we found a severe shift of protein complex "signatures", especially for the WOX5-PLT3 dimer in the QC and CSCs and BRAVO-PLT3 dimer in SIs. This further strengthens our hypothesis that the protein complexes form instructive protein signatures important for cell fate decisions in the *Arabidopsis* root SCN.

Interestingly, full-length PLT3 under the control of the *WOX5* promoter only partially rescues the *plt3-1* periclinal QC division phenotype. This emphasizes that functional PLT3 is also necessary to locally maintain CSC fate and repress differentiation as the QC divides to replenish lost CSCs (Cruz-Ramírez et al, 2013). Furthermore, this could indicate a specific function for PLT3 in the CSC fate, as the presence of other PLTs could not fully compensate for the loss of PLT3. Previous findings in yeast suggest that differences in IDRs mediate the specificity of transcription factors that share the same DNA-binding motif (Brodsky et al, 2020). This is often observed among TFs that belong to the same family. If a similar mechanism also exists in plants, this could suggest that PLT3 function in CSC fate is linked specifically to its PrDs and that, due to their differentially structured PrDs, the other PLTs cannot compensate for this specific function. These findings are supported by previous observations that truncated *pPLT3:PLT3ΔPrD-mV* in the *plt2, plt3* double mutant can rescue the QC division phenotype to a *plt2* single mutant level but only partially rescues the CSC differentiation phenotype (Burkart et al, 2022). Additionally, this could indicate that mobile PLT3, which might move from the QC to CSC, is not enough to maintain CSC stem cell character.

In the future, it should be addressed how the predicted protein complex "signatures" described here are integrated in the complex gene regulatory networks in the root SCN (Cruz-Ramírez et al, 2012; García-Gómez et al, 2017; Pardal and Heidstra, 2021) to drive

changes in gene expression, including *BRAVO*, *PLT3*, and *WOX5*, but also other target genes, and how this relates to QC division and CSC number alterations in single and multiple mutants. As a next step, the model could also consider the role of cell-cell mobility of free protein (Mähönen et al, 2014; Pi et al, 2015; García-Gómez et al, 2020; Betegón-Putze et al, 2021), the presence of membrane-less compartments to account for the localization of WOX5-PLT3 in nuclear bodies in the CSC (Burkart et al, 2022), and other key regulatory processes involved and how this relates to QC division and CSC number alterations in single and multiple mutants. The integration of experimental and computational approaches holds promise to uncover these complex mechanisms underlying root SCN maintenance.

Overall, our results suggest that BRAVO, PLT3, and WOX5 form cell-type specific profiles of protein complexes and that proper complex formation contributes to optimal stem cell maintenance. Furthermore, we propose that these unique protein complex signatures serve as a read-out for cell specificity and could explain the different roles played by BRAVO, PLT3, and WOX5 in the regulation of stem cell homeostasis in the root and overall root development.

# Methods

### Reagents and tools table

| Reagent/resource | Reference or source | Identifier or catalog number |
| --- | --- | --- |
| **Experimental models** | | |
| *Escherichia coli DB3.1* | | F– gyrA462 endA1 Δ(sr1recA) mcrB mrr hsdS20(rB–, mB) supE44 ara-14 galK2 lacY1 proA2rpsL20(SmR) xyl-5 λ– leu mtl1 |
| *Escherichia coli DH5a* | | F– Φ80lacZΔM15 Δ(lacZYA-argF) U169 recA1 endA1 hsdR17 (rK–, mK + ) phoA supE44 λ– thi1gyrA96 relA1 |
| *Agrobacterium tumefaciens* | | GV3101::pMP50 |
| *Nicotiana benthamiana* | | N/A |
| *A. thaliana/bravo-2* | Vilarrasa-Blasi et al, 2014 | SALK_062413 |
| *A. thaliana/wox5-1* | Galinha et al, 2007 | SALK_038262 |
| *A. thaliana/plt3-1* | Burkart et al, 2022 | SALK_127417 |
| *A. thaliana/Col-0, pBRAVO:BRAVO-mV* | This study | N/A |
| *A. thaliana/Col-0, pPLT3:PLT3-mV* | Burkart et al, 2022 | N/A |
| *A. thaliana/Col-0, pWOX5:WOX5-mV* | | N/A |
| *A. thaliana/bravo-2, plt3-1* | This study | N/A |
| *A. thaliana/bravo-2, wox5-1* | Betegón-Putze et al, 2021 | N/A |
| *A. thaliana/plt3-1, wox5-1* | Burkart et al, 2022 | N/A |
| *A. thaliana/bravo-2, plt3-1, wox5-1* | This study | N/A |
| *A. thaliana/plt3-1, pWOX5:GR-PLT3-mTurquoise2* | | N/A |
| *A. thaliana/plt3-1, pWOX5:GR-PLT3ΔPrD-mTurquoise2* | | N/A |

| Reagent/resource | Reference or source | Identifier or catalog number |
| --- | --- | --- |
| **Recombinant DNA** | | |
| pGGB002 | Addgene | #48820 |
| pGGD007 | | #48837 |
| pGGE009 | | #48841 |
| pGGF002 | | #48843 |
| pGGG001 | | #48850 |
| pGGG002 | | #48851 |
| pGGM000 | | #48864 |
| pGGN000 | | #48865 |
| pGGZ001 | | #48868 |
| pRD42 | Burkart et al, 2022 | pRD42 |
| pRD43 | | pRD43 |
| pRD53 | | pRD53 |
| pRD45 | | pRD45 |
| pRD40 | | pRD40 |
| pRD41 | | pRD41 |
| pRD65 | | pRD65 |
| pRD101 | | pRD101 |
| pPD161 | Denninger et al, 2019 | Ubi-XVE oLexA-35S in pGGA000 |
| pVS125 | This study | pVS125 |
| pRD135 | | pRD135 |
| pVS191 | | pVS191 |
| pVS84 | | pVS84 |
| pJM81 | Maika et al, 2023 | pJM81 |
| pJM82 | | pJM82 |
| pBLAD011 | Simon Lab | mTurquoise2 in pGGD002 |
| GreenGate expression vectors | This study | Appendix Table S4 |
| XVE:PLT3-mCherry | Burkart et al, 2022 | N/A |
| XVE:WOX5-mCherry | | N/A |
| XVE:PLT3ΔPrD-mCherry | | pRD138 |
| **Oligonucleotides and other sequence-based reagents** | | |
| Primers used for cloning | This study | Appendix Table S1 |
| Primers used for genotyping | This study | Appendix Table S2 |
| **Chemicals, enzymes, and other reagents** | | |
| Alexa Fluor™ 488 Azid (Alexa Fluor™ 488 5-Carboxamido-(6-Azidohexanyl), Bis(Triethylammoniumsalz)), 5-Isomer | Invitrogen™ | Catalog number A10266 |
| EdU (5-Ethynyl-2'-desoxyuridine) | Invitrogen™ | Catalog number A10044 |
| T4 DNA ligase (5 U/µl) | Thermo Scientific™ | Catalog number EL0011 |
| FastDigest Eco31l (IIs-class) | Thermo Scientific™ | Catalog number FD0293 |
| Propidium iodide | Invitrogen™ | Catalog number P1304MP |
| **Software** | | |
| SymPhoTime 64 | PicoQuant, https://www.picoquant.com/downloads | |
| R Studio | Posit Software | |
| ImageJ/Fiji | https://imagej.net/software/fiji/downloads | |

| Reagent/resource | Reference or source | Identifier or catalog number |
|---|---|---|
| Microsoft Office | Microsoft | |
| ZEN 3.0 (blue edition) | Carl ZEISS AG | |
| Origin 2021b | OriginLab Corporation | |
| **Other** | | |
| Confocal LSM 780 | Carl ZEISS AG | |
| Confocal LSM880 | Carl ZEISS AG | |
| Hydra Harp 400 | PicoQuant | |
| Pulsed laser diode | PicoQuant | |
| Tau-SPAD detectors | PicoQuant | |

## Plant work

All *Arabidopsis thaliana* lines used in this study were in *Col-0* background and can be found in Appendix Table S5. The *wox5-1* and *plt3-1* single mutants (Galinha et al, 2007) as well as the *bravo-2* single mutant (Vilarrasa-Blasi et al, 2014) and *bravo-2 wox5-1* double mutant (Betegón-Putze et al, 2021) were described before. The *bravo-2 plt3-1* double and *bravo-2 plt3-1 wox5-1* triple mutants were created by crossings. Homozygous F3 plants were verified by PCR using appropriate primers (Appendix Table S2). Transgenic lines were created by the floral dip method (Zhang et al, 2006). The *pPLT3:PLT3-mV* and *pWOX5:WOX5-mV* translational reporters in *Col-0* WT background were described earlier (Burkart et al, 2022). For *pBRAVO:BRAVO-mVenus*, *pWOX5:GR-PLT3-mTurquoise2*, and *pWOX5:GR-PLT3ΔPrD-mTurquoise2* homozygous transgenic plants, lines were selected, that possess a single T-DNA insertion, which was tested by observing the segregation on selection marker containing plates. Plants for crossing, genotyping, transformation, floral dip, and amplification were grown under long-day conditions (8 h dark, 16 h light) at 21 °C and 60% humidity. For microscopy, seeds were sterilized with chlorine gas (50 ml 13% sodium hypochlorite (v/v), 1 ml hydrochloric acid) in a desiccator, mounted in 0.15% (w/v) agarose and stratified in the dark at 4 °C for minimum 2 days before sowing on GM agar plates without sucrose (1/2 MS including Gamborg B5 vitamins, 1.2% plant agar (w/v) and 0.05% MES hydrate (w/v)). Seedlings for imaging were grown for 5–6 days under continuous light at 80 μmol m$^{-2}$ s$^{-1}$, 21 °C, and 60% humidity.

## Cloning

Plasmids for the transgenic lines *pBRAVO:BRAVO-mVenus*, *pWOX5:GR-PLT3-mTurquoise2* and *pWOX5:GR-PLT3ΔPrD-mTurquoise2* as well as for transient expression in *N. benthamiana* were generated using the GreenGate cloning method in the pGGZ001 destination vector (Lampropoulos et al, 2013). The region of the *WOX5* promoter, the CDS of WOX5, PLT3, and PLT3ΔPrD CDS as well as WOX5, PLT3, and PLT3ΔPrD constructs for transient expression in *N. benthamiana* were described before (Burkart et al, 2022). The region upstream of the transcriptional start of BRAVO (2925 bp) (Lee et al, 2006) was assigned as promoter and amplified by PCR with appropriate primers containing flanking *Bsa*I restriction sites and matching overlaps for GreenGate cloning. The internal *Bsa*I recognition site in the *BRAVO* promoter region was not removed, but incubation times for restriction digestion and GreenGate reaction were adapted accordingly. After PCR, the promoter sequence was cloned

into the GreenGate entry vector pGGA000 using *Bsa*I restriction and ligation. The CDS of BRAVO and TPL were amplified from cDNA derived from extracted RNA by PCR using primers carrying the *Bsa*I recognition site and matching GreenGate overhangs. Next, they were cloned into the GreenGate entry vector pGGC000 via restriction digest and ligation. All entry vectors were confirmed by sequencing. The GreenGate entry vector carrying the β-estradiol inducible promoter cassette was provided by (Denninger et al, 2019). For bimolecular fluorescence complementation, the GreenGate M and N intermediate vectors, each of which carried one expression cassette, were used. The correct assembly of the modules was confirmed by sequencing. All module combinations, constructs as well as primers used for cloning are listed in Appendix Tables S1, S3, and S4, respectively.

## SCN staining

SCN staining was performed according to (Burkart et al, 2022). For CSC layer quantification, optical longitudinal sections of the *Arabidopsis* root were acquired. The cell layer below the QC was scored as differentiated if three or more cells in this layer accumulated starch granules. QC cell divisions were quantified using an optical cross-section of the RAM on a scale of zero to four or more cells. If the QC was duplicated and showed two layers, as often seen for *bravo-2* mutants, only QC divisions in the upper layer were counted.

The CSC layer and QC cell division phenotypes were visualized separately in bar plots using Microsoft Excel (Microsoft Office 365, Microsoft Corporation). To assess potential correlations between CSC layers and QC divisions, data were combined into 2D-plots showing QC division on the *x* axis and CSC layer on the *y* axis using Origin 2021b (OriginLab Corporation).

## Root length measurements

Seedlings were grown for 10 days on GM plates without sucrose under continuous light at 80 μmol m$^{-2}$ s$^{-1}$, 21 °C and 60% humidity, before the plates were imaged (CanoScan 9000 F, Canon). Root length was quantified using Fiji (ImageJ) (Schindelin et al, 2012).

## Transient expression in *Nicotiana benthamiana*

For transient expression in *N. benthamiana*, the *Agrobacterium* strain GV3101::pMP50 was used that in addition to the plasmid harboring the desired construct, carried the helper plasmid pSOUP needed for GreenGate vectors. *Agrobacteria* were grown overnight in 5 ml dYT medium at 28 °C with shaking. After centrifugation for 10 min at 4000 rpm and 4 °C, the pellet was resuspended in infiltration medium (5% sucrose (w/v), 0.01% MgSO$_4$ (w/v), 0.01% glucose (w/v) and 450 μM acetosyringone) to an optical density OD$_{600}$ of 0.6 and mixed with an *Agrobacterium* strain carrying the p19 silencing repressor and possibly with a second *Agrobacterium* strain carrying a different construct for co-expression. Subsequently, the cultures were incubated for 1 h at 4 °C. To trigger stomatal opening and thereby allow easy infiltration, *N. benthamiana* plants were sprayed with water and kept under high humidity prior to infiltration. The abaxial side of the leaf was infiltrated using a syringe without a needle. Expression was induced 2–4 days after infiltration by spraying a 20 μM β-estradiol solution containing 0.1% Tween®-20 (v/v) to the abaxial side of the leaf. Depending on the expression level, FLIM measurements were performed 2–16 h after induction.

## Microscopy

Imaging of *Arabidopsis thaliana* roots was performed using an inverted ZEISS LSM 780 or LSM880. For cell wall staining, Arabidopsis seedlings were mounted in an aqueous solution of propidium iodide (PI) (10 µM). Fluorophores and fluorescent dyes were excited and detected as follows: PI was excited with 561 nm and detected at 590–670 nm; Alexa Fluor® 488 was excited at 488 nm and detected at 500–580 nm; mVenus was excited at 514 nm and detected at 520–570 nm, and mCherry was excited at 561 nm and detected at 580–680 nm. When mVenus was co-expressed with mCherry, it was excited at 488 nm and detected at 505–555 nm.

## Intensity measurements of protein levels in *A. thaliana*

For analysis of expression levels of different reporters in 6 DAG *Arabidopsis* roots of different genotypes, an inverted LSM880 microscope with constant settings for all reporters was used. The mean fluorescence levels were measured in ImageJ using an oval region of interest (ROI) of the size of one nucleus. One to three nuclei were measured per cell type and root of which the mean was calculated. Data were normalized to the mean value of the combination cell type and reporter that yielded the highest intensity. Data resulted from three technical replicates.

## Induction of GR inducible Arabidopsis lines

For the *plt3-1* rescue experiments, seeds were sown on GM agar plates without sucrose (1/2 MS including Gamborg B5 vitamins, 1.2% plant agar (w/v) and 0.05% MES hydrate (w/v)) containing either 0.1% DMSO (v/v) for control condition or 20 µM DEX (diluted in DMSO) for GR induction. After 5 days, seedlings were transferred to GM agar plates without sucrose containing 7 µg/ml 5-ethynyl-2'-deoxyuridine (EdU) and either 0.1% DMSO (v/v) or 20 µM DEX (diluted in DMSO) and grown for 24 h. SCN staining, imaging, and scoring of QC divisions and CSC layers were performed as described above.

## FRET-FLIM measurements

FRET-FLIM measurements were performed in transiently expressing epidermal leaf cells of 3 to 4 weeks old *N. benthamiana* using an inverted ZEISS LSM 780 equipped with additional time-correlated single-photon counting devices (Hydra Harp 400, PicoQuant GmbH) and a pulsed laser diode. mVenus was chosen as donor and excited at 485 nm with 1 µW laser power at the objective (40 x C-Apochromat/1.2 Corr W27, ZEISS) and a frequency of 32 MHz and detected using two τ-SPAD single-photon counting detectors in perpendicular and parallel orientation. Photons were collected over 40 frames at 256 × 256 pixels per frame, a pixel dwell time of 12.6 µs and a digital zoom of 8. Before image acquisition, a calibration routine was performed. Fluorescence correlation spectroscopy (FCS) measurements of deionized water and Rhodamine110 were acquired to test system functionality. Additionally, monitoring the decay of erythrosine B in saturated potassium iodide served as an instrument response function (IRF) to correct the fitting for system-specific time shift between laser pulse and data acquisition.

First, fluorescence decays of the donor-only control were analyzed using the "Grouped FLIM" analysis tool to determine the average fluorescence lifetime using a mono- or biexponential fitting model (SymPhoTime, PicoQuant GmbH). Next, to extract information about protein affinities and proximities, the "Grouped LT FRET Image" tool was utilized for a monoexponentially decaying donor, and the "One Pattern Analysis (OPA)" tool was used for samples with a biexponentially decaying donor (SymPhoTime, PicoQuant GmbH). These tools allow separate analyses of the amplitude and fluorescence lifetime of the FRET fraction of each sample. Consequently, the amplitude of the FRET component serves as a measure for the number of molecules undergoing FRET, termed binding or protein affinity, whereas the difference of the fluorescence lifetime of the FRET component compared to the lifetime of the donor-only fraction is used to calculate the FRET efficiency which serves as a measure for protein proximity and orientation (Maika et al, 2023). For samples where molecules do not undergo FRET e.g., the donor-only and negative control, binding values mostly varied between −10 and 10% and corresponding FRET efficiencies mostly accumulated at 10 or 80%, which was defined during the fitting process (Maika et al, 2023).

## Statistical tests

Data were tested for normal distribution by Shapiro test ($\alpha = 0.05$) followed by a Levene's test for equality of variances ($\alpha = 0.05$). Since some data did not show normal distribution or equality of variances or both, all datasets were tested with a nonparametric Kruskal–Wallis with post hoc Dunn's test ($\alpha = 0.05$) using Benjamini and Hochberg correction. To compare differences in periclinal QC division frequencies of DMSO or DEX-treated seedlings of different genotypes, Pearson's Chi-squared test with Yates' continuity correction was used. Similarly, periclinal QC division frequencies were compared across different genotypes, using an omnibus Pearson's Chi-squared test followed by a pairwise comparison. *P* values were adjusted after Benjamini and Hochberg (Benjamini and Hochberg, 1995). Statistical testing was performed using R.

## Protein complex modeling

To estimate the relative association and dissociation rates for each of the dimeric and trimeric complexes studied here, we used the following ordinary differential equations:

$$\frac{dDA}{dt} = a{\cdot}A{\cdot}D - d{\cdot}DA \qquad (1)$$

$$\frac{dA}{dt} = d{\cdot}DA - a{\cdot}AD \qquad (2)$$

$$\frac{dD}{dt} = d{\cdot}DA - a{\cdot}A{\cdot}D \qquad (3)$$

where *DA* is the protein complex formed by donor protein *D* and acceptor protein *A*. Using these equations, we simulated that the amount of protein complex, *DA*, is determined by the product of the association rate (*a*), the concentrations of donor, *D*, and acceptor, *A*, proteins, and how much it dissociates depends on the dissociation rate (*d*). To explain the relative binding affinity values determined experimentally for each dimeric and trimeric protein complex, we assessed association and dissociation rates involved in the protein complex formation from a

wide range of values (0–0.5 arbitrary units, step 0.0002), and simulated the protein complex *AB* formation until a steady state was reached. We deemed a particular combination of association and dissociation rates successful if they produce a value of *AB* at steady state in line with the relative binding affinity rates (allowing a 0.00001 deviation). In this way, we were able to predict relative binding rates for the dimeric and trimeric protein complexes studied here. The multiple parameter options we obtained differ in the speed at which the protein complex is formed, yet all produce the same protein complex level at steady state (Appendix Fig. S5A). Indeed, the FRET/FLIM measurements used to select for association and dissociation parameters provide us with information of the binding affinities, and not of the dynamics of protein complex formation.

Next, we simulated the protein complex formation in the cells of the root SCN using the following ordinary differential equations to describe the formation of each dimeric and trimeric complex:

$$\frac{dWOX5}{dt} = d_{BRAVOWOX5} \cdot BRAVOWOX5 + d_{WOX5PLT3} \cdot WOX5PLT3 + d_{WOX5PLT3BRAVO2} \cdot WOX5PLT3BRAVO - WOX5 \cdot (a_{BRAVOWOX5} \cdot BRAVO + a_{WOX5PLT3} \cdot PLT3 + a_{WOX5PLT3BRAVO2} \cdot BRAVOPLT3)$$

(4)

$$\frac{dBRAVO}{dt} = d_{BRAVOWOX5} \cdot BRAVOWOX5 + d_{BRAVOPLT3} \cdot BRAVOPLT3 + d_{WOX5PLT3BRAVO1} \cdot WOX5PLT3BRAVO - BRAVO \cdot (a_{BRAVOWOX5} \cdot WOX5 + a_{BRAVOPLT3} \cdot PLT3 + a_{WOX5PLT3BRAVO1} \cdot WOX5PLT3)$$

(5)

$$\frac{dPLT3}{dt} = d_{BRAVOPLT3} \cdot BRAVOPLT3 + d_{WOX5PLT3} \cdot WOX5PLT3 - PLT3 \cdot (a_{BRAVOPLT3} \cdot BRAVO + a_{WOX5PLT3} \cdot WOX5)$$

(6)

$$\frac{dWOX5PLT3}{dt} = a_{WOX5PLT3} \cdot WOX5 \cdot PLT3 - d_{WOX5PLT3} \cdot WOX5PLT3 - a_{WOX5PLT3BRAVO1} \cdot WOX5PLT3 \cdot BRAVO + d_{WOX5PLT3BRAVO1} \cdot WOX5PLT3BRAVO$$

(7)

$$\frac{dBRAVOPLT3}{dt} = a_{BRAVOPLT3} \cdot BRAVO \cdot PLT3 - d_{BRAVOPLT3} \cdot BRAVOPLT3 - a_{WOX5PLT3BRAVO2} \cdot BRAVOPLT3 \cdot WOX5 + d_{WOX5PLT3BRAVO2} \cdot WOX5PLT3BRAVO$$

(8)

$$\frac{dBRAVOWOX5}{dt} = a_{BRAVOWOX5} \cdot BRAVO \cdot WOX5 - d_{BRAVOWOX5} \cdot BRAVOWOX5$$

(9)

$$\frac{dWOX5PLT3BRAVO}{dt} = a_{WOX5PLT3BRAVO1} \cdot WOX5PLT3 \cdot BRAVO + a_{WOX5PLT3BRAVO2} \cdot BRAVOPLT3 \cdot WOX5 - WOX5PLT3BRAVO \cdot (d_{WOX5PLT3BRAVO1} + d_{WOX5PLT3BRAVO2})$$

(10)

The trimeric complex can be formed either by the binding of BRAVO to WOX5-PLT3, or WOX5 to BRAVO-PLT3. Then, we modeled the protein complexes formed in the cells of the root SCN using Eqs. (4)–(10) and the following relative protein levels of WOX5, BRAVO, and PLT3 determined for SI, QC, CSC, and CC cells as initial condition (Appendix Table S21). Importantly, we are using the experimentally determined protein levels in each cell as a

constant and with our modeling approach we study how these are partitioned into different protein complexes based on the binding parameters we determined in silico. As several sets of binding rates were predicted per complex, for these simulations we used one selected at random. The differences in the parameters are related to the speed at which the protein complexes are formed, yet we have no dynamic information to assess which one is the correct one. Notably, the specific parameters used for the results we present here do not markedly change the protein complex signatures predicted by the model (Appendix Fig. S5B). Moreover, we tested how the output of our simulations changes when all parameter describing protein complex formation are either in the fast (highest association parameter predicted) or the slow range (lowest association parameter predicted). As can be seen in Appendix Fig. S8 in both cases the output of the simulations is remarkably similar, thus supporting our approach of focusing on the analysis of the steady states reached.

To evaluate the effect in our model of both, the cell-type specific protein levels as well as differential binding affinities are necessary for our model, we performed different control simulations. On the one hand, we tested the effect of equal association/dissociation rates ($a = d = 0.1$), higher association than dissociation rate ($a = 0.1$, $d = 0.05$), and lower association than dissociation ($a = 0.05$, $d = 0.1$) for all protein complexes using our experimental protein level quantification in the SI, QC, CSC, and CC displayed as Control 1–3, respectively (Appendix Fig. S6). On the other hand, we consider an alternative scenario where all proteins have the same abundance levels, while the association/dissociation rates are based on our binding data (Control 4, Appendix Fig. S6). Finally, we consider the scenarios where the control conditions meet pairwise: Control 5 is a combination of equal association/dissociation rate together with the assumption of equal protein abundances among cell types and proteins. In Control 6, the equality of protein levels is combined with higher association than dissociation rates. Finally, Control 7 combines lower association that dissociation rates with equal protein abundances (Appendix Fig. S6). Notably, only control 2, which uses experimentally determined protein abundances together with a higher association than dissociation rate, produced results comparable to our model. Thus, leading to the conclusion that also in our experimental data, association rates must be higher than dissociation rates. Moreover, this indicates a key role of the protein levels in each cell in the resulting protein complex and free protein signatures. In all other cases, we could observe strikingly different protein complex "signatures" to those we described with the model that uses our experimental data, indicating that our findings result from the combination of experimentally determined specific protein levels and binding affinities.

## Data availability

The datasets and computer code produced in this study are available in the following databases: Imaging datasets: BIA: https://www.ebi.ac.uk/biostudies/bioimages/studies/S-BIAD1636?key=1a3406df-caa0-483b-ba74-93faa36b0b24; Computational modeling: (https://tbb.bio.uu.nl/monica/Protein-complexes-root_SCN), GitHub (https://github.com/moneralee/Cell-type-specific-complex-formation-of-key-transcription-factors-in-the-root_SCN) or Zenodo (Garcia Gomez, 2025).

The source data of this paper are collected in the following database record: biostudies:S-SCDT-10_1038-S44319-025-00422-8.

## Peer review information

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

## Acknowledgements

Large parts of the work and the manuscript are based on the published PhD thesis of VIS (Heinrich-Heine University, Düsseldorf). We would like to acknowledge funding of VIS by the Deutsche Forschungsgemeinschaft (DFG) through grant 432468382 to YS. MLGG is supported by the long-term program PlantXR: A new generation of breeding tools for extra-resilient crops (KICH3.LTP.20.005) which is financed by the Dutch Research Council (NWO), the Foundation for Food & Agriculture Research (FFAR), companies in the plant breeding and processing industry, and Dutch universities. These parties collaborate in the CropXR Institute (www.cropxr.org) that is funded through the National Growth Fund (NGF) of the Netherlands. We thank Rebecca C. Burkart for sharing PLT3, PLT3ΔPrD, and WOX5 constructs and stable Arabidopsis lines. The authors thank Ana Caño-Delgado for sharing seeds of bravo-2 single and bravo-2 wox5-1 double mutants. The authors thank Kirsten ten Tusscher for insightful discussions on the modeling and Jan Kees van Amerongen for the management of computational facilities of the Theoretical Biology group (Utrecht University). The authors thank Cornelia Gieseler, Carin Theres and Silke Winters for technical assistance. The authors thank Meik H. Thiele for help with statistical analyses and data visualization with R. The authors also thank Jan E Maika for help with fitting FRET-FLIM data. The authors would like to acknowledge the Center for Advanced Imaging (CAi) at Heinrich-Heine-University Düsseldorf for providing access to the Zeiss LSM 780 and LSM 880 and especially Dr. Sebastian Hänsch and Prof. Dr. Stefanie Weidtkamp-Peters for general support during imaging and analysis. Funding for instrumentation: Zeiss LSM 780: DFG- INST 208/551-1 FUGG and Zeiss LSM880: DFG- INST 208/746-1 FUGG. Furthermore, the authors thank Ksenia Krooß (NFDI4BIOIMAGE, DFG project ID 501864659) for help with metadata annotation and upload to the BioImage Archive repository.

## Author contributions

**Vivien I Strotmann**: Conceptualization; Data curation; Formal analysis; Validation; Investigation; Visualization; Methodology; Writing—original draft; Writing—review and editing. **Monica L Garcia-Gomez**: Conceptualization; Formal analysis; Funding acquisition; Visualization; Methodology; Writing—original draft; Writing—review and editing. **Yvonne Stahl**: Conceptualization; Data curation; Formal analysis; Supervision; Funding acquisition; Validation; Methodology; Writing—original draft; Project administration; Writing—review and editing.

Source data underlying figure panels in this paper may have individual authorship assigned. Where available, figure panel/source data authorship is listed in the following database record: biostudies:S-SCDT-10_1038-S44319-025-00422-8.

## Funding

## Disclosure and competing interests statement

The authors declare no competing interests.

# Expanded View Figures

---

**Figure EV1.  Elevated QC division frequencies negatively correlate to the number of CSC layers.**

(**A–H**) 2D histograms visualizing the combined results of the SCN staining in the respective genotype showing the number of CSC layers on the y axis and QC divisions on the x axis. Darker colors correspond to a higher number of roots showing the phenotype. Numbers of analyzed roots per genotype (biological replicates) are indicated in each graph and result from 3 to 5 technical replicates. (**I**) Close-up of the QC in the Col-0 WT. Scale bar: 5 µm. (**J**) Close-up view of the QC of a *bravo-2* mutant showing an additional periclinal cell division plane (white arrowhead). Scale bar: 5 µm. (**K**) Quantification of periclinal cell division planes in the different mutants. Correlations between periclinal QC divisions and genotype were tested with omnibus Pearson's Chi-squared test followed by pairwise comparisons with *P* value adjustment after Benjamini and Hochberg.

▶

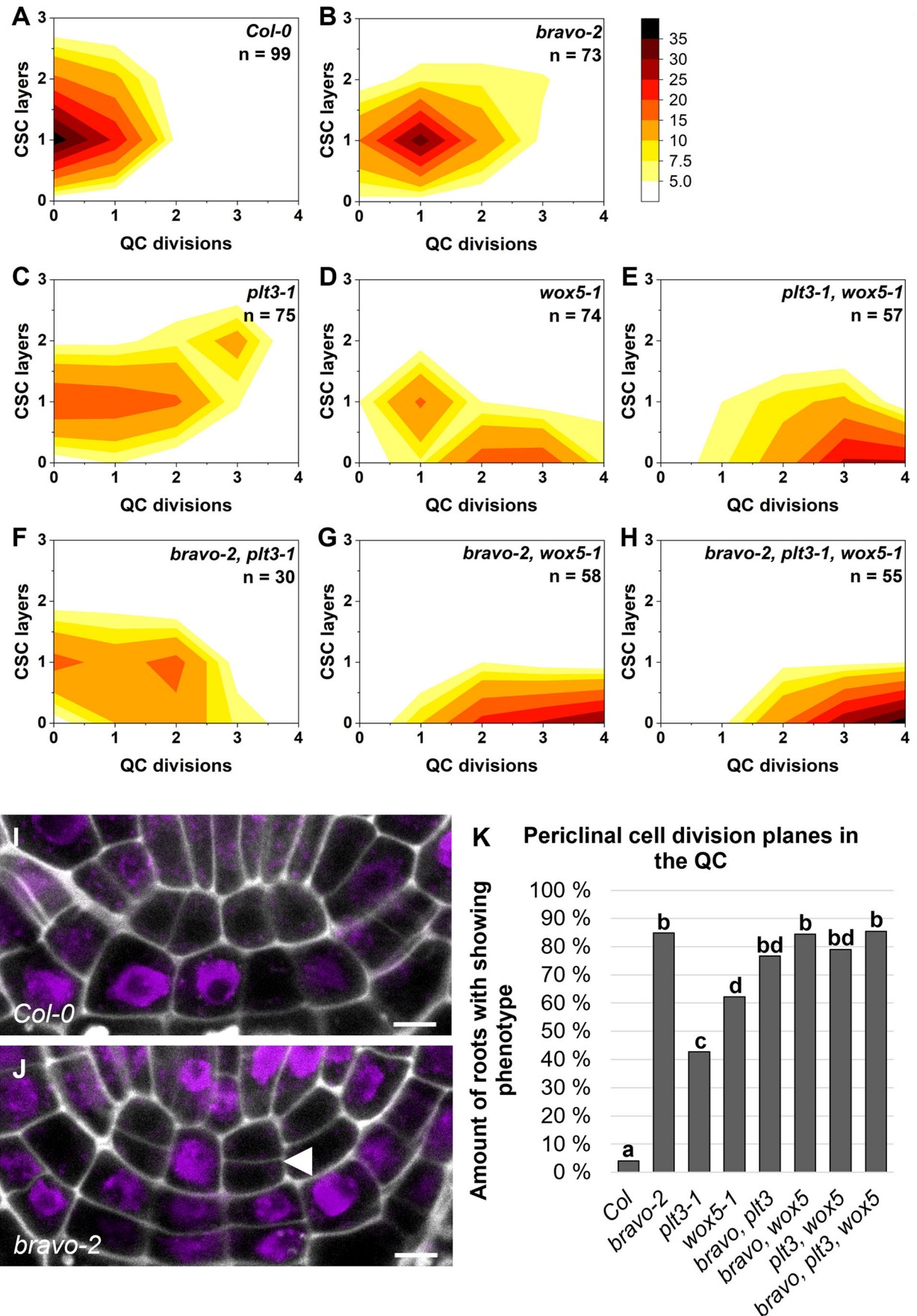

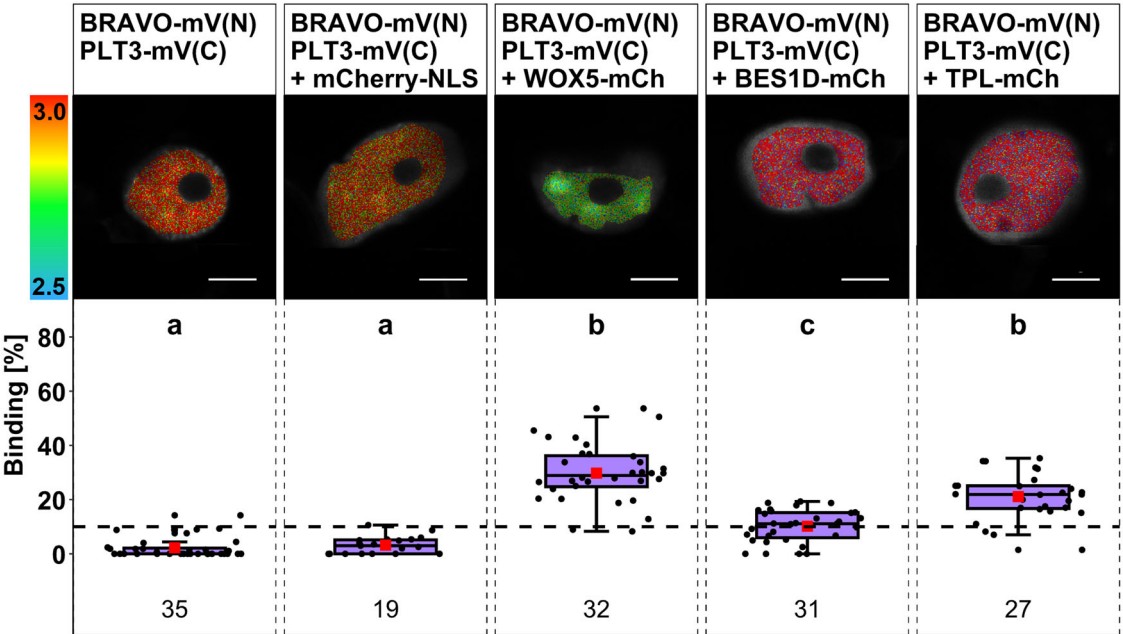

**Figure EV2.  Trimeric complex formation of BRAVO and PLT3 with WOX5, BES1D and TPL.**

Upper panel: Representative images of fluorescence lifetime imaging microscopy (FLIM) measurements in *N. benthamiana* epidermal leaf cells after a pixel-wise multiexponential fit. The fluorescence lifetime of the donor BRAVO-mV(N) PLT3-mV(C) in the presence or absence of the indicated acceptor is color-coded: blue (2.5) refers to low fluorescence lifetime [in ns], red (3.0) indicates high fluorescence lifetime. Scale bars: 6 μm. Lower panel: Binding [%] (magenta) for BRAVO-mV(N) PLT3-mV(C) with or without co-expression of mCherry-NLS, WOX5-mCh, BES1D-mCh or TPL-mCh. Statistical groups were assigned after a nonparametric Kruskal–Wallis with post hoc Dunn's test (α = 0.05, *p* values were adjusted after Benjamini and Hochberg). The black dotted line indicates the Binding cutoff of 10%. The numbers of analyzed nuclei (biological replicates) are indicated below each sample and result from 3-4 technical replicates. Data Information: Box = middle 50% of data ( = interquartile range (IQR)); whiskers = from IQR to min/max values, but at most 1.5 * IQR; line within box = median; data beyond the end of whiskers are ‚outliers' and are plotted independently. In addition, a jitter plot was used to visualize all data points individually.

## 1. Prediction of association/dissociation for complexes

### a. Experimentally determined binding affinity

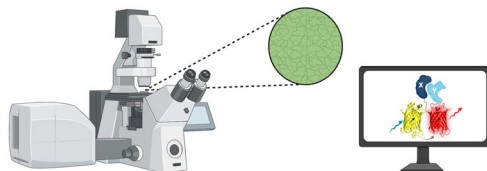

### b. Define association (a) and dissociation (b) rates

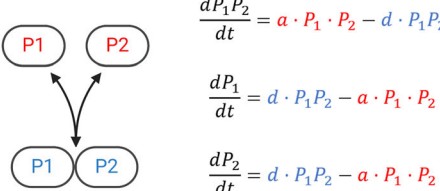

$$\frac{dP_1P_2}{dt} = a \cdot P_1 \cdot P_2 - d \cdot P_1P_2$$

$$\frac{dP_1}{dt} = d \cdot P_1P_2 - a \cdot P_1 \cdot P_2$$

$$\frac{dP_2}{dt} = d \cdot P_1P_2 - a \cdot P_1 \cdot P_2$$

### c. Evaluate outcome

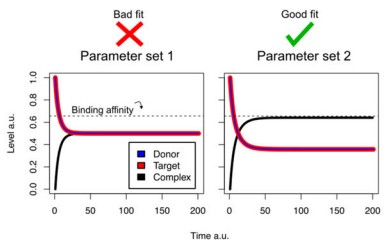

## 2. Simulation of protein complex formation in the root SCN

### a. Combine with experimentally determined protein abundances in the different cell types

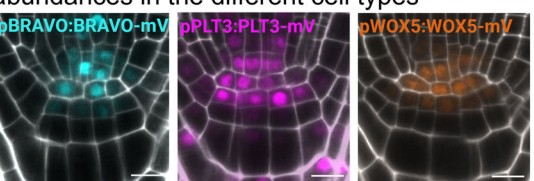

### b. Simulate cell type specific protein complex formation

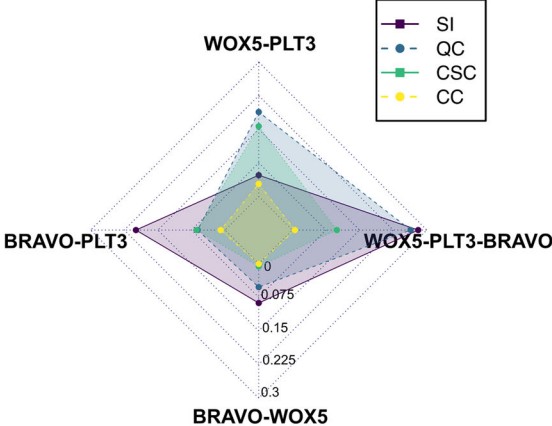

**Figure EV3.  Workflow of the mathematical modeling.**

1a: First, we experimentally determined the binding affinity of all TF combinations. **1b**: For the simulation of protein complex formation in the root, we first defined association (a) and dissociation (d) rates for each combination of TFs and saved them as a parameter set. **1c**: Each parameter set was evaluated whether it can reproduce the experimentally determined binding affinities from our FRET-FLIM studies. If so, the parameter set was kept. For each combination, many different parameter sets were able to reproduce the experimental data. **2a, b**: Finally, one parameter set was randomly chosen and combined with the experimentally determined protein abundances to finally simulate the cell-type specific protein complex formation.

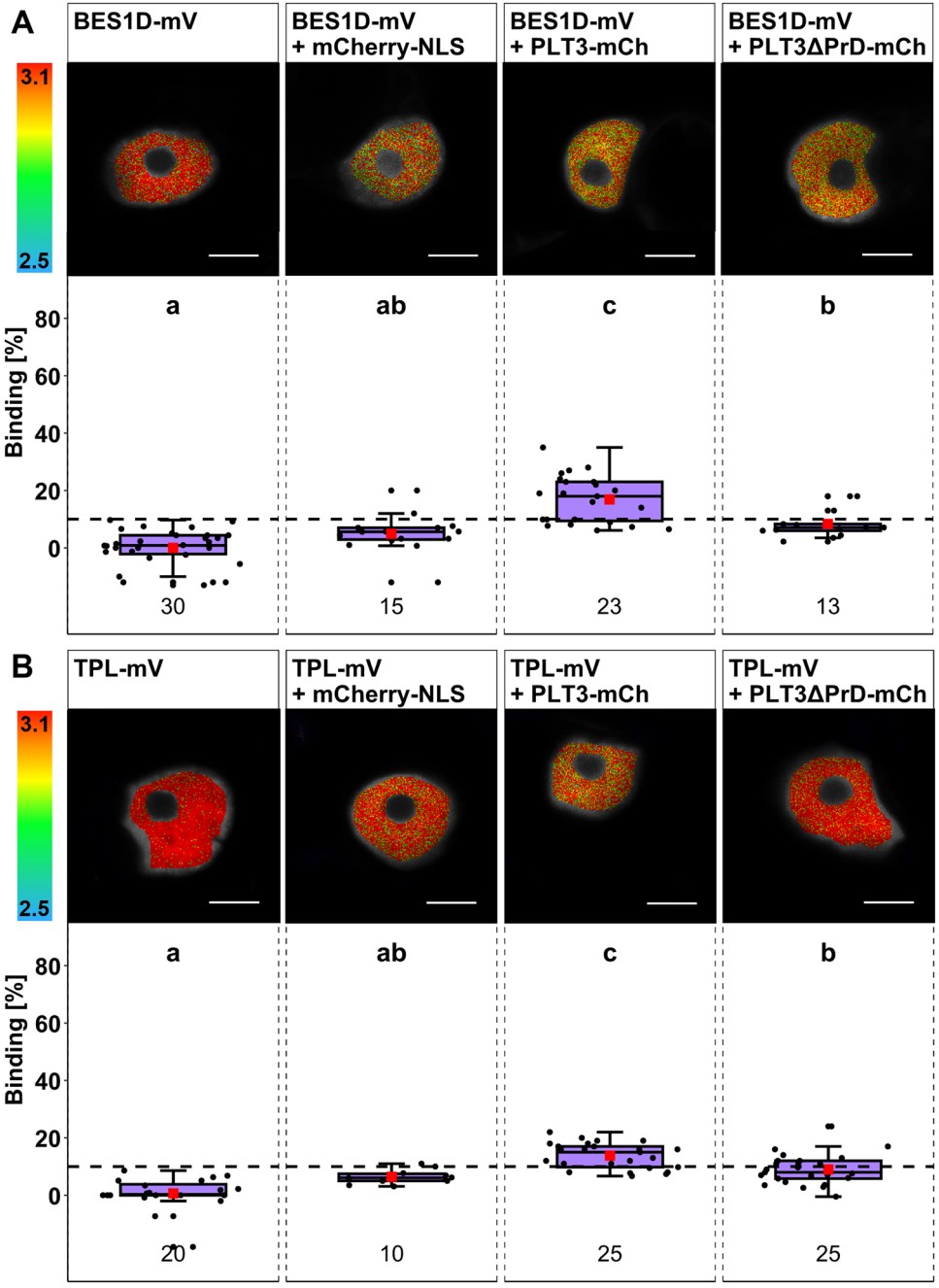

**Figure EV4. Interaction of PLT3 with BES1D and TPL depends on PrDs found in PLT3.**

(A) Upper panel: Representative images of fluorescence lifetime imaging microscopy (FLIM) measurements in *N. benthamiana* epidermal leaf cells after a pixel-wise multiexponential fit. The fluorescence lifetime of the donor BES1D-mV in the presence or absence of the indicated acceptor is color-coded: blue (2.5) refers to low fluorescence lifetime [in ns], red (3.1) indicates high fluorescence lifetime. Scale bar represents 6 μm. (A) Lower panel: Binding [%] (magenta) for BES1D-mV with or without co-expression of mCherry-NLS, PLT3-mCh or PLT3ΔPrD-mCh. Statistical groups were assigned after a nonparametric Kruskal–Wallis with post hoc Dunn's test (α = 0.05, *p* values were adjusted after Benjamini and Hochberg). The black dotted line indicates the Binding cutoff of 10%. The numbers of analyzed nuclei (biological replicates) are indicated below each sample and result from 3 technical replicates. (B) Upper panel: Representative images of FLIM measurements in *N. benthamiana* epidermal leaf cells after a pixel-wise multiexponential fit. The fluorescence lifetime of the donor TPL-mV in the presence or absence of the indicated acceptor is color-coded: blue (2.5) refers to low fluorescence lifetime [in ns], red (3.1) indicates high fluorescence lifetime. Scale bar represents 6 μm. (B) Lower panel: Binding [%] (magenta) for TPL-mV in the absence or presence of mCherry-NLS, PLT3-mCh or PLT3ΔPrD-mCh. Statistical groups were assigned after a nonparametric Kruskal–Wallis with post hoc Dunn's test (α = 0.05, *P* values were adjusted after Benjamini and Hochberg). The black dotted line indicates the Binding cutoff of 10%. The numbers of analyzed nuclei (biological replicates) are indicated below each sample and result from 2 technical replicates. Data information: (A, B) Box = middle 50% of data ( = interquartile range (IQR)); whiskers = from IQR to min/max values, but at most 1.5 * IQR; line within box = median; data beyond the end of whiskers are "outliers" and are plotted independently. In addition, a jitter plot was used to visualize all data points individually.

