## [Peer Review File · EMBO Reports]

Root stem cell homeostasis in Arabidopsis involves cell type specific transcription factor complexes

Vivien Strotmann, Monica Garcia-Gomez, and Yvonne Stahl

Corresponding author: Yvonne Stahl (y.stahl@bio.uni-frankfurt.de)

Review Timeline:

Transferred from Review Commons:	11th Dec 24
Editorial Decision:	20th Jan 25
Revision Received:	17th Feb 25
Editorial Decision:	26th Feb 25
Revision Received:	4th Mar 25
Accepted:	6th Mar 25

Editor: Achim Breiling

Transaction Report:

This article was transferred to EMBO Reports following peer review at Review Commons.

Review #1**1. Evidence, reproducibility and clarity:****Evidence, reproducibility and clarity (Required)**

The manuscript "Stem cell homeostasis in the root of Arabidopsis involves cell type specific complex formation of key transcription factors" presents a detailed analysis of the transcription factors (TFs) BRAVO, PLT3, and WOX5 in maintaining stem cell homeostasis in Arabidopsis thaliana. The authors mapped TF abundances in different cell types within the stem cell niche (SCN), created triple mutants to demonstrate their combined effects on quiescent center (QC) divisions, and used Fluorescence Resonance Energy Transfer Fluorescence Lifetime Imaging Microscopy (FRET-FLIM) to quantify protein-protein interactions. Their computational modeling incorporated in vivo binding affinities and abundances, suggesting that differential dimer and trimer formation of these TFs contributes to SCN regulation. Notably, the interaction between BRAVO and PLT3 and the trimeric complex BRAVO-PLT3-WOX5 are novel findings. These findings deepen our understanding of these well-known TFs in the SCN.

****Major comments:****

1. The authors used native promoter lines in a wild-type background to measure protein abundances. Confirming whether these native promoters complement the mutants and are expressed at physiological levels is essential, as the insertion site could affect expression.
2. The manuscript introduces a novel method for calculating binding affinities from FRET-FLIM measurements. It would be beneficial to compare these binding affinities with those measured in vitro to validate the approach.
3. The experiments in Figure 7 show limited rescue of the *plt3* mutant phenotype by PLT3 driven by the WOX5 promoter. It would be useful to test if these constructs rescue the mutant phenotype when expressed under the native PLT3 promoter.
4. The computational model provides testable predictions, such as the potential formation of the trimeric complex without the PLT Prion Domain (PrD). This could be experimentally verified using the assays described by the authors.

****Minor comments:****

1. The rationale for setting the association rate of PLT3 Δ PrD-WOX5 to zero in Figure 6 needs clarification. Is the interaction entirely abolished?

2. The discussion is lengthy and could be streamlined to enhance the clarity of the manuscript.

2. Significance:

Significance (Required)

The study presents significant advances in understanding the regulation of stem cell homeostasis in Arabidopsis roots. The strongest aspects include the detailed mapping of TF abundances, the novel interactions discovered, and the integration of experimental and computational approaches. However, the study could benefit from further validation of key findings and a clearer explanation of certain experimental choices.

The research will be of broad interest to plant biologists, particularly those studying stem cell regulation and transcription factor interactions. The findings have implications beyond Arabidopsis, potentially informing studies in other plant systems and broader developmental biology contexts.

My expertise includes stem cell regulation, transcription factor dynamics, and plant developmental biology. While I am familiar with most aspects of the paper, I would recommend additional input from a specialist in computational modeling for a thorough evaluation of the modeling approaches used.

3. How much time do you estimate the authors will need to complete the suggested revisions:

Estimated time to Complete Revisions (Required)

(Decision Recommendation)

Between 1 and 3 months

4. Review Commons values the work of reviewers and encourages them to get credit for their work. Select 'Yes' below to register your reviewing activity at Web of Science Reviewer Recognition Service (formerly Publons); note that the content of your review will not be visible on Web of Science.

Yes

Review #2

1. Evidence, reproducibility and clarity:

Evidence, reproducibility and clarity (Required)

****Summary****

In this study, Strotmann et al build further on their previous report of a PLT3-WOX5 complex regulating root stem-cell maintenance (Burkart et al 2022). Using in vivo live imaging the authors create a cell-type specific profile of the three TFs BRAVO, PLT3, and WOX5 and establish a cell-specific stoichiometry of the three proteins, which is required for maintaining the architecture of the stele initials (SIs), quiescent centre (QC), and the columella stem cells (CSCs). The FRET-FLIM experiments combined with BiFC provide evidence to the formation of a novel BRAVO-PLT3 dimer along with potential higher-order trimeric complexes between BRAVO-PLT3-WOX5. Using the above datasets, the authors prepare a computational model which defines the relative free and bound levels of WOX5-PLT3 in the QC and CSCs, and PLT3-BRAVO in the SIs as essential regulators of root meristem cell numbers and types. The study is easy to follow and adds detailed nuance to the importance of PLT3 and BRAVO alongside WOX5 for meristem homeostasis. The experiments are well thought-out with no superfluous inferences derived. Since this reviewer does not have a background in computational modelling, there are only a few minor suggestions for improving the clarity of the data.

Fig 1., Lines 134-138. Can the authors explain if the individual transgenic lines were determined to be equally stable based on transgene copy number, transcript levels or homozygosity? The authors have stated that the BRAVO, PLT3, and WOX5 mVenus transgenic lines are stable, there is no supplementary information provided about any transgene copy numbers or transcript levels to support the claim.

Fig 1. Would the age of the plant make a difference to the expression domains of BRAVO and PLT3? Have the authors checked this? It would be interesting to see if the protein abundance of the complexes changes during development.

Fig 6. It would be helpful if the authors can add some schematics to show the prion-like domains of PLT3, so as to provide clarity on what region(s) has been deleted to disrupt complex formation.

Lines 448-450: in Dex treated *plt3-1 pWOX5:iPLT3* roots the reduction of additional QC

divisions does not seem so significant, 87% to 67% may not be a definitive rescue of the extra QC divisions especially if not complemented with either BRAVO or WOX5 in this scenario. The authors should discuss for the possibility of some non-cell autonomous effects of PLT3 and its interactors from the SIs and CSCs on the QC itself.

2. Significance:

Significance (Required)

The study provides useful nuance to the established repertoire of TFs essential for root meristem maintenance. Building on previous studies, the authors combine live imaging, biochemical protein-protein interaction measurements, with computational modelling to build a detailed framework of individual TF stoichiometry within each stem-cell type in the root. This will serve as a useful tool to derive observations on different types of TF interactions within the stem cell niche required for development of individual cell-layers. While the complexes studied here have been presently defined to each stem-cell type, further questions arise on the dynamic nature of these complexes under different conditions.

3. How much time do you estimate the authors will need to complete the suggested revisions:

Estimated time to Complete Revisions (Required)

(Decision Recommendation)

Between 1 and 3 months

Yes

Review #3

1. Evidence, reproducibility and clarity:

Evidence, reproducibility and clarity (Required)

In this interesting manuscript Strotman and colleagues focuses on the protein-protein interactions that occurs in a cells specific manner and that are crucial for stem cells maintenance. They mainly focus on three different transcription factors (BRAVO, PLT3 and WOX5) that were known to interact between them. Via genetic and biochemical experiments they show that WOX5 interact with PLT3 and that PLT3 can interact with BRAVO to maintain stem cells. Via a combination of wet and in silico biology they show that dimers of PLT3 and BRAVO define sieve initials whereas dimers of WOX5 and PLT3 define QC and CSC. They identify the PLT3 ID domain as a crucial domain for protein-protein interaction and in particular for interaction with BRAVO. Exploiting this knowledge they also show that PLT3-BRAVO interaction is fundamental for PLT3 activity in the SCN.

The manuscript is well written and rational is well explained. Results are consistent and performed with rigor. Results are well discussed, despite the discussion section is very long and repetitive.

Despite the manuscript report several interesting results it is not clear to this reviewer how the reported interactions affects SCN maintenance and, hence, general meristem function. Indeed, *wox5* mutants defects are known to alter the overall SCN maintenance as mutations in *WOX5* enhances *shr* and *scr* phenotype, causing additional malfunctioning of the QC and accelerating meristem consumption. It is not clear to me at what degree PLT3 and BRAVO alters the overall activity of SCN as single mutants do not exhaust it and combinations with *wox5* mutants are not reported in this optic. Authors should analyse the root development of their single mutant and mutant combinations. I would suggest to focus especially on *plt3*, *bravo* and single mutant plants with altered number of periclinal division.

Also to permit readers to understand the degree of involvement of these genes in the regulation of the QC, authors should analyse QC markers that are missing in *wox5* mutants such as QC184 in mutant backgrounds. These experiments will be crucial to assess how the proposed protein-protein interactions alters SCN maintenance.

Also it is not clear when authors refers to periclinal divisions, authors should explain better in the introduction what those are , what they cause and what they represent in the general SCN maintenance.

In this manuscript and in previous manuscripts authors reported that PLT3 is able to interact with WOX5 and that ID promoter is important for PLT3 function in SCN maintenance. As expression of *WUS* in the QC is sufficient to recover *wox5* defects and

PLT3 is fundamental for shoot function and regeneration authors should discuss whether this interaction might be crucial also for WUS activity.

Also to test their predictions I would suggest authors to eliminate in a tissue specific manner their interactors. These experiments will allow to unveil how alteration in specific cells alters the SCN maintenance.

****Minor points:****

- Authors should report number of plants analysed
- Authors should clearly state age of analysed plants in figure legends
- Fig 1J and T: statistical treatments are missing.

2. Significance:

Significance (Required)

This manuscript will be of large interest for scientists working on stem cell field. Indeed beyond the novelty reported in root development, it propose new ideas on how protein-protein interactions can affect stem cell activity in a cell specific manner in multicellular organisms. In particular the combination of in vivo and in silico methodologies to analyze protein-protein interaction might be of wide interest. Nonetheless, the functions of these interactions in vivo must be strengthened.

3. How much time do you estimate the authors will need to complete the suggested revisions:

Estimated time to Complete Revisions (Required)

(Decision Recommendation)

Between 1 and 3 months

4. Review Commons values the work of reviewers and encourages them to get credit for their work. Select 'Yes' below to register your reviewing activity at Web of Science Reviewer Recognition Service (formerly Publons); note that the content of your review will not be visible on Web of Science.

Yes

Full Revision

Manuscript number: RC-2024-02499

Corresponding author(s): Yvonne, Stahl

[Please use this template only if the submitted manuscript should be considered by the affiliate journal as a full revision in response to the points raised by the reviewers.]

*If you wish to submit a preliminary revision with a revision plan, please use our "Revision Plan" template. **It is important to use the appropriate template to clearly inform the editors of your intentions.**]*

1. General Statements [optional]

Dear referees,

we would like to thank all three referees for taking the time to give us valuable and constructive comments on our manuscript. We have carefully revised our manuscript addressing all questions raised by the referees. We have included the respective explanations to the main text and materials and methods and added comments indicating which question was answered.

Below you may find our point-by-point response to all raised questions highlighted in blue.

With kind regards on behalf of all authors

Yvonne Stahl

This section is mandatory. Please insert a point-by-point reply describing the revisions that were already carried out and included in the transferred manuscript.

Reviewer #1 (Evidence, reproducibility and clarity (Required)):

The manuscript "Stem cell homeostasis in the root of Arabidopsis involves cell type specific complex formation of key transcription factors" presents a detailed analysis of the transcription factors (TFs) BRAVO, PLT3, and WOX5 in maintaining stem cell homeostasis in Arabidopsis thaliana. The authors mapped TF abundances in different cell types within the stem cell niche (SCN), created triple mutants to demonstrate their combined effects on quiescent center (QC) divisions, and used Fluorescence Resonance Energy Transfer Fluorescence Lifetime Imaging Microscopy (FRET-FLIM) to quantify protein-protein interactions. Their computational modeling incorporated in vivo binding affinities and abundances, suggesting that differential dimer and trimer formation of these TFs contributes to SCN regulation. Notably, the interaction between BRAVO and PLT3 and the trimeric complex BRAVO-PLT3-WOX5 are novel findings. These findings deepen our understanding of these well-known TFs in the SCN.

Major comments:

Q1: The authors used native promoter lines in a wild-type background to measure protein abundances. Confirming whether these native promoters complement the mutants and are expressed at physiological levels is essential, as the insertion site could affect expression. We thank the reviewer for bringing this point to our attention. C-terminal fusions of PLT3 and WOX5 with mVenus under control of their respective native promoter have previously been shown to complement the observed CSC differentiation and QC division phenotypes (Burkart *et al.*, 2022). Similarly, we could confirm that the C-terminally tagged BRAVO under control of its endogenous promoter (Lee *et al.*, 2006) is able to rescue the observed SCN phenotypes. We have added this information to the main text and refer to an additional supplementary figure S1 in lines 129-134: *‘[...] previously described pPLT3:PLT3-mV and pWOX5:WOX5-mV homozygous translational reporters in Col-0 WT background. Both reporters have been shown to complement their respective mutant phenotype (Burkart et al., 2022). Additionally, we generated a homozygous, stable transgenic Arabidopsis line expressing pBRAVO: BRAVO-mV also in the Col-0 WT background, which is capable to complement the bravo-2 mutant phenotype (Fig. S1).’*

Q2: The manuscript introduces a novel method for calculating binding affinities from FRET-FLIM measurements. It would be beneficial to compare these binding affinities with those measured in vitro to validate the approach.

We thank the referee for raising this important point. The fitting routine used to calculate the binding affinities from the FRET-FLIM data, was introduced and established before in different publications (Orthaus *et al.*, 2009; Maika *et al.*, 2023). Here, binding is introduced as the relative amplitude of the FRET fraction of the fluorescence decay of the donor fluorophore and

describes the relative number of molecules that undergo FRET within a sample. This allows us to deduce differences in protein affinities of different TFs.

To our knowledge, *in vitro* measurements are not available for the analysed combinations investigated in this study. Furthermore, using a combination of experimental data obtained from an orthogonal system and mathematical modelling, allowing us to simulate protein complexes in different cell types of the root SCN, seems to be more promising system. Such results with similar spatial resolution would be challenging to obtain from *in vitro* measurements. To further clarify the combination of experimental data and computational modelling, we added a supplementary figure that describes the workflow of the mathematical modelling approach (see Fig. S9) and added a statement to the main text (lines 336-338) '*To enhance clarity on the workflow of our mathematical modelling approach, we included a supplementary figure that summarizes the fundamental steps of this process (Fig. S9).*'

Q3: The experiments in Figure 7 show limited rescue of the *plt3* mutant phenotype by PLT3 driven by the WOX5 promoter. It would be useful to test if these constructs rescue the mutant phenotype when expressed under the native PLT3 promoter.

Previous observations show that PLT3 C-terminally fused to mVenus under control of its own promoter is able to complement the observed SCN defects in *plt3-1* single mutants (Burkart *et al.*, 2022). Furthermore, the same construct rescues the *plt2*, *plt3* double mutant phenotype to *plt2* single mutant levels. However, truncated PLT3 Δ PrD C-terminally fused to mVenus under control of the *PLT3* promoter in the *plt2*, *plt3* double mutant background leads to an intermediate rescue phenotype: on the one hand, truncated PLT3 Δ PrD and full-length PLT3 possess a similar ability to rescue the QC division phenotype, partially rescuing the *plt2*, *plt3* double mutant to the *plt2* single mutant levels. On the other hand, the ability to rescue CSC layer phenotype is more pronounced using full-length PLT3 compared to the truncated PLT3 Δ PrD. Therefore, we indicated that '*This emphasizes that functional PLT3 is also necessary to locally maintain CSCs fate and repress differentiation as the QC divides to replenish lost CSCs (Cruz-Ramírez et al., 2013)*' (line 620-622) and that PrDs possess an additional specific role in cell fate determination in CSCs. To connect our results with previous findings, we further added '*These findings are supported by previous observations that truncated pPLT3:PLT3 Δ PrD-mV in the *plt2*, *plt3* double mutant is able to rescue the QC division phenotype to a *plt2* single mutant level but only partially rescues the CSC differentiation phenotype (Burkart et al., 2022).*' (line 630-633) to the main text.

Q4: The computational model provides testable predictions, such as the potential formation of the trimeric complex without the PLT Prion Domain (PrD). This could be experimentally verified using the assays described by the authors.

We thank the reviewer for this valuable suggestion. To verify the predictions displayed in Fig. 5 A, C and Fig. 7 J, L, we performed FRET-FLIM measurements of WOX5-mV(N) and BRAVO-mV(C) in the presence of PLT3-mCh or PLT3 Δ PrD-mCh, respectively (Fig. S11, supplementary table S20). We found that the trimeric complex WOX5-BRAVO-PLT3 shows an average binding of 31.5 ± 8.5 % and the complex WOX5-BRAVO-PLT3 Δ PrD a not significantly different average

binding of 28.9 ± 8.4 %, which is in accordance with our simulations showing a minor decrease in the abundance of the trimeric complex WOX5-PLT3-BRAVO in the absence of PLT3 PrDs in SIs and the QC (Fig. 7J, L). This important information was also added to the main text *'Surprisingly, the abundance of the trimeric complex in SIs and the QC was only mildly affected. Here, in line with our modelling approach, we could experimentally verify by FRET-FLIM measurements of WOX5-mV(N) and BRAVO-mV(C) in the presence of either PLT3-mCh or PLT3ΔPrD-mCh, yielding an average binding of 31.5 ± 8.5 % and 28.9 ± 8.4 %, respectively (Fig. S11).'* (lines 491-495).

Minor comments:

Q5: The rationale for setting the association rate of PLT3ΔPrD-WOX5 to zero in Figure 6 needs clarification. Is the interaction entirely abolished?

We thank the reviewer for bringing this to our attention. We have decided to reassess the published data to extract distinct binding values from previous FLIM-FRET experiments (Burkart *et al.*, 2022). Here, similarly to the interaction of BRAVO and PLT3, we found that the interaction of PLT3 and WOX5 is lowered by the loss of PLT3 PrDs but not completely abolished (Fig. S5). Therefore, we changed the respective paragraphs in the main text (lines 421-425) *'Furthermore, we reassessed the published data on WOX5 interaction with PLT3 and PLT3ΔPrD (Burkart et al., 2022) revealing that interaction of WOX5 and PLT3 is not completely abolished upon loss of PrDs but significantly reduced from 30.6 ± 13.3 % with full-length PLT3 to 18.5 ± 12.4 % in the absence of PLT3 PrDs (Fig. S5).'* Additionally, we added the full results of our analysis as supplementary Fig. S5 and updated the correlating supplementary table S15. Moreover, we repeated the computational simulations of PLT3ΔPrD using the updated binding affinity with WOX5 (i.e. 18.5 ± 12.4 %). Due to this change in protein affinity, we noticed slight changes in the predicted cell type specific complex formation (Fig. 5). Namely, in SIs and the QC, the trimeric complex is the most abundant protein complex. Before, SIs were characterized by high levels of BRAVO-PLT3 dimer and the QC by the WOX5-PLT3 dimer. High levels of WOX5-PLT3 dimer in CSCs did not change. We adapted the respective sections in the main text as follows: (lines 375-380) *'Our simulation reveals that SIs and QC cells are characterized by high levels of the trimeric protein complex WOX5-PLT3-BRAVO (Fig. 5 A, C). The CSC cells are predicted to be enriched in the WOX5-PLT3 complex. Such enrichment could be related to the previously described function of the WOX5-PLT3 complex in CSC maintenance (Burkart et al., 2022).'* as well as lines 485-495 *'The reduced affinity of PLT3ΔPrD towards WOX5 and BRAVO alters dimer formation causing a redistribution of BRAVO and WOX5 to the other protein complexes and an increase of free BRAVO, PLT3 and WOX5 protein levels in nearly all simulated cell types (Fig. 7 J, K). While the BRAVO-PLT3 and WOX5-PLT3 complexes can still be formed, their levels are noticeably reduced in the SI, QC and CSC cells. Furthermore, the BRAVO-WOX5 complex levels increase in the SI and QC cells. Surprisingly, the abundance of the trimeric complex in SIs and the QC is only mildly affected. Here, in line with our modelling approach, we could experimentally verify by FRET-FLIM measurements of WOX5-mV(N) and BRAVO-mV(C) in the presence of either PLT3-mCh or PLT3ΔPrD-mCh, yielding an average binding of 31.5 ± 8.5 % and 28.9 ± 8.4 %, respectively (Fig. S11).'*

Full Revision

Furthermore, we also adapted parts of our discussion: lines 554-566 '*Namely, high levels of the WOX5-BRAVO-PLT3 trimer appear to be characteristic for stele initials and QC cells, which can be further discriminated by high free BRAVO protein abundance in the SIs and high free WOX5 levels in the QC (Fig. 1). Several studies highlighted the elevated abundance of WOX5 in the QC, which could be either linked to interactions with other proteins not analysed here or its non-cell autonomous function in the adjacent initials, although its necessity as mobile stemness factor is still under debate (Pi et al., 2015; Berckmans et al., 2020). The CSCs were predicted to be enriched in the WOX5-PLT3 heterodimer, which aligns with their previously described impact on CSC differentiation (Burkart et al., 2022). Furthermore, CSC possess high levels of PLT3 (Fig. 5), which might be linked to nuclear body (NB) formation of PLT3, which was linked to its PrDs and is concentration dependent and may involve PLT3 homomerization.*' as well as lines 578-582 '*Our findings suggest that the additional pathways involve the combinations BRAVO and PLT3, as well as BRAVO and WOX5 and the formation of a trimeric complex. This indicates that these TFs could act in four independent constellations to regulate SCN maintenance depending on which protein they are partnering with, [...]*'.

Due to these changes in cell type specific complexes, we had to adapt Fig. 5, Fig. 8, Fig. S5, Fig. S6 and Fig. S7 accordingly.

Q6: The discussion is lengthy and could be streamlined to enhance the clarity of the manuscript. We thank the reviewer for this valuable suggestion. We adapted the discussion to include answers to comments from other reviewer as well as shortened some points to enhance the clarity of the discussion.

Reviewer #1 (Significance (Required)):

The study presents significant advances in understanding the regulation of stem cell homeostasis in Arabidopsis roots. The strongest aspects include the detailed mapping of TF abundances, the novel interactions discovered, and the integration of experimental and computational approaches. However, the study could benefit from further validation of key findings and a clearer explanation of certain experimental choices.

The research will be of broad interest to plant biologists, particularly those studying stem cell regulation and transcription factor interactions. The findings have implications beyond Arabidopsis, potentially informing studies in other plant systems and broader developmental biology contexts.

My expertise includes stem cell regulation, transcription factor dynamics, and plant developmental biology. While I am familiar with most aspects of the paper, I would recommend additional input from a specialist in computational modeling for a thorough evaluation of the modeling approaches used.

Reviewer #2 (Evidence, reproducibility and clarity (Required)):

Evidence, reproducibility, and clarity

SUMMARY

In this study, Strotmann et al build further on their previous report of a PLT3-WOX5 complex regulating root stem-cell maintenance (Burkart et al 2022). Using in vivo live imaging the authors create a cell-type specific profile of the three TFs BRAVO, PLT3, and WOX5 and establish a cell-specific stoichiometry of the three proteins, which is required for maintaining the architecture of the stele initials (SIs), quiescent centre (QC), and the columella stem cells (CSCs). The FRET-FLIM experiments combined with BiFC provide evidence to the formation of a novel BRAVO-PLT3 dimer along with potential higher-order trimeric complexes between BRAVO-PLT3-WOX5. Using the above datasets, the authors prepare a computational model which defines the relative free and bound levels of WOX5-PLT3 in the QC and CSCs, and PLT3-BRAVO in the SIs as essential regulators of root meristem cell numbers and types. The study is easy to follow and adds detailed nuance to the importance of PLT3 and BRAVO alongside WOX5 for meristem homeostasis. The experiments are well thought-out with no superfluous inferences derived. Since this reviewer does not have a background in computational modelling, there are only a few minor suggestions for improving the clarity of the data.

Q1: Fig 1., Lines 134-138. Can the authors explain if the individual transgenic lines were determined to be equally stable based on transgene copy number, transcript levels or homozygosity? The authors have stated that the BRAVO, PLT3, and WOX5 mVenus transgenic lines are stable, there is no supplementary information provided about any transgene copy numbers or transcript levels to support the claim.

We thank the reviewer for this important question. We have clarified our statement in the results part as follows: *'[...] by measuring the fluorescence intensity of mVenus (mV) in nuclei of the previously described pPLT3:PLT3-mV and pWOX5:WOX5-mV homozygous translational reporters in Col-0 WT background. Both reporters have been shown to complement their respective mutant phenotype (Burkart et al., 2022). Additionally, we generated a homozygous, stable transgenic Arabidopsis line expressing pBRAVO:BRAVO-mV also in the Col-0 WT background, which is capable to complement the bravo-2 mutant phenotype (Fig. S1)'* (lines 128-134), emphasizing that the used lines are homozygous. Additionally, for further information we have referred to the Material and Methods section, where we have stated in more detail, how the transgenic lines were generated see lines 665-670: *'The pPLT3:PLT3-mV and pWOX5:WOX5-mV translational reporters in Col-0 WT background were described earlier (Burkart et al., 2022). For pBRAVO:BRAVO-mVenus, pWOX5:GR-PLT3-mTurquoise2, and pWOX5:GR-PLT3 Δ PrD-mTurquoise2 homozygous transgenic plants, lines were selected, that possess a single T-DNA insertion, which was tested by observing the segregation on selection marker containing plates.'*

Q2: Fig 1. Would the age of the plant make a difference to the expression domains of BRAVO and PLT3? Have the authors checked this? It would be interesting to see if the protein abundance of the complexes changes during development.

We thank the reviewer for this interesting question. So far, our experiments cover protein complex abundances corresponding to BRAVO, PLT3 and WOX5 protein availability at 5 DAG. We have chosen to monitor plants at this stage of development for our experiments since 5 DAG marks the time point where post-embryonic root organization and the size of the meristem has fully established (Moubayidin *et al.*, 2010). In the future, the mathematical modelling approach offers the opportunity to investigate changes in protein complex abundances over time by implementing protein availabilities at earlier or later developmental stages, e.g. 3, 7 or 10 DAG. However, we decided to limit the scope of this study to one time point and save temporal changes for further research.

Q3: Fig 6. It would be helpful if the authors can add some schematics to show the prion-like domains of PLT3, so as to provide clarity on what region(s) has been deleted to disrupt complex formation.

We thank the referee for this suggestion. To enhance clarity on what regions of PLT3 have been replaced or deleted, we have added a scheme of PLT3 aa sequence, including known domains, to Figure 6.

Q4: Lines 448-450: in Dex treated *plt3-1* pWOX5:iPLT3 roots the reduction of additional QC divisions does not seem so significant, 87% to 67% may not be a definitive rescue of the extra QC divisions especially if not complemented with either BRAVO or WOX5 in this scenario. The authors should discuss for the possibility of some non-cell autonomous effects of PLT3 and its interactors from the SIs and CSCs on the QC itself.

We thank the reviewer for this important observation. While the reduction of additional QC divisions in DEX-treated *plt3-1* pWOX5:iPLT3 roots (from 87% to 67%) may not represent a complete rescue, this partial effect suggests that PLT3 alone is not fully sufficient to restore normal QC behaviour. We agree that this points to the possibility of non-cell-autonomous effects of PLT3 and its interactors from surrounding tissues, such as the stele initials (SIs) and columella stem cells (CSCs), influencing QC regulation. This aligns with the growing understanding that QC maintenance, as it is crucial for the plant, is orchestrated by a network of dynamic signals across different cell types to build a plant typical strong compensation mechanism. Here, the cell-type specific interactions between PLT3 and other factors like BRAVO or WOX5 that we are reporting may be necessary for complete QC regulation. We have included a short discussion of this potential mechanism in the revised manuscript lines 627-635 '[...] *If a similar mechanism also exists in plants, this could suggest that PLT3 function in CSC fate is specifically linked to its PrDs and that, due to their differentially structured PrDs, the other PLTs cannot compensate for this specific function. These findings are supported by previous observations that truncated pPLT3:PLT3ΔPrD-mV in the plt2, plt3 double mutant is able to rescue the QC division phenotype to a plt2 single mutant level but only partially rescues the CSC differentiation phenotype (Burkart et al., 2022). Additionally, this could indicate that mobile*

Full Revision

PLT3 which might move from the QC to CSC is not enough to maintain CSC stem cell character.'

Reviewer #2 (Significance (Required)):

Significance

The study provides useful nuance to the established repertoire of TFs essential for root meristem maintenance. Building on previous studies, the authors combine live imaging, biochemical protein-protein interaction measurements, with computational modelling to build a detailed framework of individual TF stoichiometry within each stem-cell type in the root. This will serve as a useful tool to derive observations on different types of TF interactions within the stem cell niche required for development of individual cell-layers. While the complexes studied here have been presently defined to each stem-cell type, further questions arise on the dynamic nature of these complexes under different conditions.

Full Revision

Reviewer #3 (Evidence, reproducibility and clarity (Required)):

In this interesting manuscript Strotman and colleagues focuses on the protein-protein interactions that occurs in a cells specific manner and that are crucial for stem cells maintenance. They mainly focus on three different transcription factors (BRAVO, PLT3 and WOX5) that were known to interact between them. Via genetic and biochemical experiments they show that WOX5 interact with PLT3 and that PLT3 can interact with BRAVO to maintain stem cells. Via a combination of wet and in silico biology they show that dimers of PLT3 and BRAVO define sieve initials whereas dimers of WOX5 and PLT3 define QC and CSC. They identify the PLT3 ID domain as a crucial domain for protein-protein interaction and in particular for interaction with BRAVO. Exploiting this knowledge they also show that PLT3-BRAVO interaction is fundamental for PLT3 activity in the SCN.

Q1: The manuscript is well written and rational is well explained. Results are consistent and performed with rigor. Results are well discussed, despite the discussion section is very long and repetitive.

We thank the reviewer for this valid suggestion. We adapted the discussion to include answers to comments from other reviewer as well as shortened some points to enhance the clarity of the discussion.

Q2: Despite the manuscript report several interesting results it is not clear to this reviewer how the reported interactions affects SCN maintenance and, hence, general meristem function. Indeed, *wox5* mutants defects are known to alter the overall SCN maintenance as mutations in WOX5 enhances *shr* and *scr* phenotype, causing additional malfunctioning of the QC and accelerating meristem consumption. It is not clear to me at what degree PLT3 and BRAVO alters the overall activity of SCN as single mutants do not exhaust it and combinations with *wox5* mutants are not reported in this optic. Authors should analyse the root development of their single mutant and mutant combinations. I would suggest to focus especially on *plt3,bravo* and single mutant plants with altered number of periclinal division.

We thank the reviewer for this important question. We have conducted experiments for single and multiple mutants to integrate the observed altered number of periclinal divisions with bigger scale impacts on root length for biological function. Here we found significantly shorter roots in the *bravo plt3 wox5* triple mutant compared to the *bravo plt3* double and *bravo-2* single mutant, which possess slightly yet not significantly longer roots compared to the other analysed mutants and the wildtype. These results could strengthen and extend our hypothesis and the results of the modelled protein complex formation in the root SCN: Loss of any of these TFs causes a redistribution of protein complexes. However, up to a certain degree, the root can compensate for potential losses, either promoting cell elongation in the elongation zone or cell divisions in the meristem, or by other closely related TFs, or by partially overlapping functions of BRAVO, PLT3 and WOX5 in the root SCN. We have added our findings to the main text (lines 229-234) *'To further investigate the consequences of SCN defects occurring at 5 DAG on the overall root development, we analysed the total primary root length of these mutants at 10 DAG (Fig. S3). Here we found that, in comparison to the bravo plt3 double mutants the bravo plt3 wox5 triple mutant roots are significantly shorter, once more highlighting the importance and combinatory*

effect of BRAVO, PLT3 and especially WOX5 for proper root meristem maintenance.' and added this data into the supplementary figure S3.

Q3: Also to permit readers to understand the degree of involvement of these genes in the regulation of the QC, authors should analyse QC markers that are missing in *wox5* mutants such as QC184 in mutant backgrounds. These experiments will be crucial to assess how the proposed protein-protein interactions alters SCN maintenance.

We thank the reviewer for this valuable suggestion. We did not incorporate enhancer trap markers such as the QC184 marker into our analyses, as their functional characterization and underlying mechanisms remain incomplete. Instead, we have included additional data on the effects of the various single and multiple mutants on root length, which provide in our opinion more informative insights into the biological function of these mutations, as suggested by the reviewer (see above, Q2).

Q4: Also it is not clear when authors refers to periclinal divisions, authors should explain better in the introduction what those are , what they cause and what they represent in the general SCN maintenance.

We thank the referee for this suggestion. We have adapted the introduction by providing more information on periclinal QC divisions in lines 59-62 '*[...] first it produces the surrounding tissue-specific stem cells, also referred to as initials, by formative, asymmetric cell divisions, which occur either anticlinally or periclinally. These stem cells divide asymmetrically to give rise to different cell types[...]*' and lines 64-66 '*The QC mostly divides periclinally thereby producing predominantly columella stem cells (CSC) (Cruz-Ramírez et al., 2013).*'

Q5: In this manuscript and in previous manuscripts authors reported that PLT3 is able to interact with WOX5 and that ID promoter is important for PLT3 function in SCN maintenance. As expression of WUS in the QC is sufficient to recover *wox5* defects and PLT3 is fundamental for shoot function and regeneration authors should discuss whether this interaction might be crucial also for WUS activity.

We thank the reviewer for this insightful suggestion. While this is indeed an important aspect worth investigating in a follow-up study, it falls outside the specific focus of this paper, which is centered on the root meristem. We have also opted not to include a statement on this topic in the discussion, as it was already significantly condensed at the request of the reviewers.

Q6: Also to test their predictions I would suggest authors to eliminate in a tissue specific manner their interactors. These experiments will allow to unveil how alteration in specific cells alters the SCN maintenance.

We thank the reviewer for this thoughtful suggestion. While eliminating the interactors in a tissue-specific manner would indeed provide valuable insights, such experiments are not straightforward in this case due to the dynamic nature of TFs under investigation. WOX5 (Pi *et al.*, 2015; Berckmans *et al.*, 2020; Clark *et al.*, 2020) and PLTs (Mähönen *et al.*, 2014) are known to move between cells, which makes it technically challenging to restrict their elimination to specific tissues without affecting surrounding areas. Developing the tools and methods to

Full Revision

achieve this tissue-specific manipulation would require significant effort and falls beyond the current scope of this study. Nonetheless, this would be an important development for future research.

Minor points:

Q7: Authors should report number of plants analysed

The number of analysed plants has been added to the figure legends.

Q8: Authors should clearly state age of analysed plants in figure legends

The age of analysed plants has been added to the figure legends.

Q9: Fig 1J and T: statistical treatments are missing.

We thank the reviewer for this valid suggestion. We added the results of statistical tests to the figure and adapted the text accordingly. Additionally, we also applied statistical tests to Figure 1 B, D, F and H as well as to Fig. 7 I, Fig. S1 and Fig. S2. Details about which tests have been used can be found in the respective figure descriptions and the material and methods section.

Reviewer #3 (Significance (Required)):

This manuscript will be of large interest for scientists working on stem cell field. Indeed beyond the novelty reported in root development, it propose new ideas on how protein-protein interactions can affect stem cell activity in a cell specific manner in multicellular organisms. In particular the combination of in vivo and in silico methodologies to analyze protein-protein interaction might be of wide interest. Nonetheless, the functions of these interactions in vivo must be strengthened.

References

- Berckmans B, Kirschner G, Gerlitz N, Stadler R, Simon R.** 2020. CLE40 Signaling Regulates Root Stem Cell Fate. *Plant physiology* **182**, 1776–1792.
- Burkart RC, Strotmann VI, Kirschner GK et al.** 2022. PLETHORA-WOX5 interaction and subnuclear localization control Arabidopsis root stem cell maintenance. *EMBO reports* **23**, e54105.
- Clark NM, Fisher AP, Berckmans B et al.** 2020. Protein complex stoichiometry and expression dynamics of transcription factors modulate stem cell division. *Proceedings of the National Academy of Sciences of the United States of America* **117**, 15332–15342.
- Cruz-Ramírez A, Díaz-Triviño S, Wachsman G et al.** 2013. A SCARECROW-RETINOBLASTOMA protein network controls protective quiescence in the Arabidopsis root stem cell organizer. *PLoS biology* **11**, e1001724.
- Lee J-Y, Colinas J, Wang JY, Mace D, Ohler U, Benfey PN.** 2006. Transcriptional and posttranscriptional regulation of transcription factor expression in Arabidopsis roots. *Proceedings of the National Academy of Sciences of the United States of America* **103**, 6055–6060.
- Mähönen AP, Tusscher K ten, Siligato R et al.** 2014. PLETHORA gradient formation mechanism separates auxin responses. *Nature* **515**, 125–129.
- Maika JE, Krämer B, Strotmann VI et al.** 2023. One pattern analysis (OPA) for the quantitative determination of protein interactions in plant cells. *Plant methods* **19**, 73.
- Moubaydin L, Perilli S, Dello Iorio R, Di Mambro R, Costantino P, Sabatini S.** 2010. The rate of cell differentiation controls the Arabidopsis root meristem growth phase. *Current biology CB* **20**, 1138–1143.
- Orthaus S, Buschmann V, Bäter A, Fore S, König M, Erdmann R.** 2009. Quantitative in vivo imaging of molecular distances using FLIM-FRET.
- Pi L, Aichinger E, van der Graaff E et al.** 2015. Organizer-Derived WOX5 Signal Maintains Root Columella Stem Cells through Chromatin-Mediated Repression of CDF4 Expression. *Developmental cell* **33**, 576–588.

Dear Prof. Stahl,

Thank you for the transfer of your revised manuscript from Review Commons to EMBO reports. I have now received the reports from two of the three referees that I asked to re-evaluate the study, you will find below. Original referee #3 was completely unresponsive to my invitations to re-assess the study. However, going through your p-b-p-response, I consider his/her points as adequately addressed.

As you will see, referees #1 and #2 now fully support the publication of the study in EMBO reports. Referee #1 has some remaining points and suggestions to improve the manuscript I invite you to address in a final revised manuscript.

Moreover, during a previous submission to a different EMBO press journal an advisor identified a potential error in one of the modeling equations and suggested changes. I think these points also need to be addressed. Please find also his/her report below.

Please provide a final p-b-p-response with your revised manuscript addressing the remaining concerns of referee #1 and the points of the advisor.

Finally, the manuscript needs formatting according to our journal style. Please carefully review the instructions that follow below.

When submitting your final revised manuscript, we will require:

1) a .docx formatted version of the final manuscript text (including legends for main figures, EV figures and tables), but without the figures included. Figure legends should be compiled at the end of the manuscript text.

2) individual production quality figure files as .eps, .tif, .jpg (one file per figure), of main figures and EV figures. Please upload these as separate, individual files upon re-submission.

The Expanded View format, which will be displayed in the main HTML of the paper in a collapsible format, has replaced the Supplementary information. You can submit up to 5 images as Expanded View. I would thus suggest combining the present supplementary figures to have 5 final figure files. Please follow the nomenclature Figure EV1, Figure EV2 etc. The figure legend for these should be included in the main manuscript document file in a section called Expanded View Figure Legends after the main Figure Legends section. Additional Supplementary material should be supplied as a single pdf file labeled Appendix. The Appendix should have page numbers and needs to include a table of content on the first page (with page numbers) and legends for all content. Please follow the nomenclature Appendix Figure Sx, Appendix Table Sx etc. throughout the text, and also label the figures and tables according to this nomenclature.

3) a complete author checklist, which you can download from our author guidelines

(<https://www.embopress.org/page/journal/14693178/authorguide>). Please insert page numbers in the checklist to indicate where the requested information can be found in the manuscript. The completed author checklist will also be part of the RPF.

4) that primary datasets produced in this study (e.g. RNA-seq, ChIP-seq, structural and array data) are deposited in an appropriate public database. If no primary datasets have been deposited, please also state this in a dedicated section (e.g. 'No primary datasets have been generated and deposited'), see below.

The accession numbers and database should be listed in a formal "Data Availability" section (placed after Materials & Methods) that follows the model below. This is now mandatory (like the COI statement). Please note that the Data Availability Section is restricted to new primary data that are part of this study. This section is mandatory. As indicated above, if no primary datasets have been deposited, please state this in this section

Data availability

5) We now request the publication of original source data with the aim of making primary data more accessible and transparent to the reader. Our source data coordinator will contact you to discuss which figure panels we would need source data for and will also provide you with helpful tips on how to upload and organize the files.

6) Our journal encourages inclusion of *data citations in the reference list* to directly cite datasets that were re-used and obtained from public databases. Data citations in the article text are distinct from normal bibliographical citations and should directly link to the database records from which the data can be accessed. In the main text, data citations are formatted as follows: "Data ref: Smith et al, 2001" or "Data ref: NCBI Sequence Read Archive PRJNA342805, 2017". In the Reference list, data citations must be labeled with "[DATASET]". A data reference must provide the database name, accession number/identifiers and a resolvable link to the landing page from which the data can be accessed at the end of the reference. Further instructions are available at: <http://www.embopress.org/page/journal/14693178/authorguide#referencesformat>

7) Regarding data quantification and statistics, please make sure that the number "n" for how many independent experiments were performed, their nature (biological versus technical replicates), the bars and error bars (e.g. SEM, SD) and the test used to calculate p-values is indicated in the respective figure legends (also for potential EV and Appendix figures). Please also check that all the p-values are explained in the legend, and that these fit to those shown in the figure. Please provide statistical testing where applicable. Please avoid the phrase 'independent experiment', but clearly state if these were biological or technical replicates. Please also indicate (e.g. with n.s.) if testing was performed, but the differences are not significant. In case n=2, please show the data as separate datapoints without error bars and statistics. See also: <http://www.embopress.org/page/journal/14693178/authorguide#statisticalanalysis>

Please add to each legend (main and EV figures) a 'Data Information' section explaining the statistics used or providing information regarding replicates and scales.

8) Please add scale bars of similar style and thickness to microscopic images, using clearly visible black or white bars (depending on the background). Please place these in the lower right corner of the images themselves. Please do not write on or near the bars in the image but define the size in the respective figure legend.

9) Please also note our reference format:

10) We updated our journal's competing interests policy in January 2022 and request authors to consider both actual and perceived competing interests. Please review the policy <https://www.embopress.org/competing-interests> and update your competing interests if necessary. Please name this section 'Disclosure and Competing Interests Statement' and put it after the Acknowledgements section.

11) We now use CRediT to specify the contributions of each author in the journal submission system. CRediT replaces the author contribution section. Please use the free text box to provide more detailed descriptions and do NOT add an author contributions section to the manuscript text file. See also guide to authors: <https://www.embopress.org/page/journal/14693178/authorguide#authorshipguidelines>

12) Please reduce the number of keywords to five and order the manuscript sections like this, using these names: Title page - Abstract - Keywords - Introduction - Results - Discussion - Methods - Data availability section - Acknowledgements - Disclosure and Competing Interests Statement - References - Figure legends - Expanded View Figure legends

13) All Materials and Methods need to be described in the main text using our 'Structured Methods' format, which is required for all research articles. According to this format, the Materials and Methods section should include a Reagents and Tools Table (listing key reagents, primers used, experimental models, software, and relevant equipment and including their sources and relevant identifiers), uploaded as separate file, followed by a Methods and Protocols section in which we encourage the authors to describe their methods using a step-by-step protocol format with bullet points, to facilitate the adoption of the methodologies across labs. More information on how to adhere to this format as well as downloadable templates (.doc) for the Reagents and Tools Table can be found in our author guidelines (section 'Structured Methods'):

14) Please enter all the funding information also into our submission system during resubmission and make sure this is complete and similar to the one mentioned in the acknowledgements section of the manuscript text file.

15) Please mention in the acknowledgements section that large parts of the work and the manuscript text are based on the published PhD-thesis of Vivien I. Strotmann (University of Düsseldorf).

Finally, I would need from you:

- a short, two-sentence summary of the manuscript (not more than 35 words).
- three to four short (!) one sentence bullet points highlighting the key findings of your study.
- a schematic summary figure (synopsis image) in jpeg or tiff format with the exact width of 550 pixels and a height of not more than 400 pixels that can be used as a visual synopsis on our website.

I look forward to seeing a revised version of your manuscript when it is ready. Please let me know if you have questions or comments regarding the revision.

Yours sincerely,

Referee #1 (RC Referee #2):

This study investigates the role of transcription factors (TFs) BRAVO, PLT3, and WOX5 in the regulation of the stem cell niche (SCN) in *Arabidopsis thaliana*. Using a combination of mutant phenotypic analyses, advanced microscopy (FRET-FLIM), and computational modelling the authors analyse how these three transcription factors make higher-order complexes control the stem cell niche. It is interesting to see the importance of PLT3's prion-like domains which can act as a potential conserved interaction hub, in regulation of SCN maintenance and replenishment. I really like the modelling part.

1)- The authors should be careful in their interpretation of experiments where they induce PLT3 or its variant under the WOX5 promoter and examine their abilities to rescue stem cell defects in *plt3* mutants. Here PLT3 and its variant of proteins are fused with GR domain and there is no readout such as fluorescent reporter that can assess the levels of PLT3/delPLT3-GR in the nucleus. This reviewer understands that it may not be trivial to assess the protein levels in the nucleus. So the best way is to be careful while interpreting the results of these experiments.

2)- I find Some sentences long and complex, which makes it harder to follow for readers like me. I would suggest editing the text.

3)- The authors should focus on properly referencing the literature.

4)- Also, in several places grammar needs some improvement.

Referee #2 (RC Referee #1):

The authors have addressed my concerns from the previous version of this manuscript. The additional experiments, such as the FRET-FLIM validation of the computational model's predictions, strengthen the findings. The authors have clarified key points, particularly around native promoter lines and the modeling approach. They also updated the discussion to improve the clarity and focus of the paper. The study provides mechanistic insights into how transcription factor complexes regulate stem cell homeostasis in a tissue-specific manner, which is conceptually novel and broadly relevant to developmental biology. Given the combination of quantitative live imaging, genetic analysis, and computational modeling, this work offers insights that will appeal to the interdisciplinary audience of EMBO Reports. I support its publication.

Advisor:

Concerning the robustness and appropriateness of the modelling approach, the methodology is well-targeted to the experimental system in question. The ODE simulation approach while simple does generate results that are well-integrated into the experimental data, and which provide important conceptual understanding. However, I do have some suggestions:

The right-hand side of equation 8 contains an error in one of the degradation terms. The authors need to carefully check whether this is just a typo or whether the actual simulations have been performed incorrectly.

The simulation methodology finds parameter combinations that are consistent with the experimental binding affinities. However, this only constrains the ratio of parameter values (consistent with the straight lines in Fig S6). Later on in the pipeline, a random parameter pair is picked for the actual simulations of cell type specific protein complex formation. The reason for this degeneracy needs to be stated. Furthermore, there may be parameter combinations that although consistent with steady-state are actually implausible because they will affect the dynamical approach to steady-state and may cause the steady-state to be reached too slowly. These possibilities should be discussed.

The authors need to spell out where the variability for cells of a given type in the simulation in Fig S7 comes from, bearing in mind that this is a deterministic simulation. Presumably it comes from the differing dynamics: slow dynamics will cause a complex forming pathway with faster dynamics to become more important thereby altering the protein complex levels?

Referee #1 (RC Referee #2):

This study investigates the role of transcription factors (TFs) BRAVO, PLT3, and WOX5 in the regulation of the stem cell niche (SCN) in *Arabidopsis thaliana*. Using a combination of mutant phenotypic analyses, advanced microscopy (FRET-FLIM), and computational modelling the authors analyse how these three transcription factors make higher-order complexes control the stem cell niche. It is interesting to see the importance of PLT3's prion-like domains which can act as a potential conserved interaction hub, in regulation of SCN maintenance and replenishment. I really like the modelling part.

Q 1: The authors should be careful in their interpretation of experiments where they induce PLT3 or its variant under the WOX5 promoter and examine their abilities to rescue stem cell defects in *plt3* mutants. Here PLT3 and its variant of proteins are fused with GR domain and there is no readout such as fluorescent reporter that can assess the levels of PLT3/delPLT3-GR in the nucleus. This reviewer understands that it may not be trivial to assess the protein levels in the nucleus. So the best way is to be careful while interpreting the results of these experiments.

We thank the referee for this feedback. We carefully reviewed the respective paragraphs in the results (lines 477 and 481) and in the discussion sections (lines 616-619).

Q 2: I find Some sentences long and complex, which makes it harder to follow for readers like me. I would suggest editing the text.

We thank the reviewer for this valid suggestion. We have shortened and/or split long and complex sentences to enhance clarity.

Q 3: The authors should focus on properly referencing the literature.

We thank the reviewer for bringing this to our attention. We carefully reviewed the used references and adapted them where necessary or added missing references.

Q 4: Also, in several places grammar needs some improvement.

We thank the referee for this comment. We carefully revised our manuscript and improved grammar if necessary.

Advisor:

Q 1: Concerning the robustness and appropriateness of the modelling approach, the methodology is well-targeted to the experimental system in question. The ODE simulation approach while simple does generate results that are well-integrated into the experimental data, and which provide important conceptual understanding. However, I do have some suggestions:

The right-hand side of equation 8 contains an error in one of the degradation terms. The authors need to carefully check whether this is just a typo or whether the actual simulations have been performed incorrectly.

We thank the reviewer for the positive comments regarding our modelling approach to study protein complex formation in the root stem cell niche. Regarding equation 8, there is indeed a mistake in the negative term describing the association of the BRAVOPLT3 complex with WOX5 to form the trimeric protein complex WOX5PLT3BRAVO. We have corrected this mistake in the revised version of our manuscript by replacing " $a_{WOX5PLT3BRAVO2} \cdot WOX5PLT3 \cdot BRAVO$ " with " $a_{WOX5PLT3BRAVO2} \cdot BRAVOPLT3 \cdot WOX5$ " (line 871). Importantly, this mistake was only present in the Methods section of the manuscript and not in the R code we used for the simulations, and thus does not change the results we presented before.

It is worth noting, that the mentioned mistake in equation 8 is not related to protein degradation as here we are not modelling the production and degradation of WOX5, PLT3 or BRAVO, but their association and dissociation of protein complexes. Indeed, we use their experimentally determined protein levels in each root stem cell as a constant, and with the ODEs we model how these are partitioned into different protein complexes.

Q: 2 The simulation methodology finds parameter combinations that are consistent with the experimental binding affinities. However, this only constrains the ratio of parameter values (consistent with the straight lines in Fig S6). Later on in the pipeline, a random parameter pair is picked for the actual simulations of cell type specific protein complex formation. The reason for this degeneracy needs to be stated. Furthermore, there may be parameter combinations that although consistent with steady-state are actually implausible because they will affect the dynamical approach to steady-state and may cause the steady-state to be reached too slowly. These possibilities should be discussed.

The authors need to spell out where the variability for cells of a given type in the simulation in Fig S7 comes from, bearing in mind that this is a deterministic simulation. Presumably it comes from the differing dynamics: slow dynamics will case a complex forming pathway with faster dynamics to become more important thereby altering the protein complex levels?

Our parameter estimation analysis indeed recovers many combinations of association and dissociation parameters that can produce the experimental binding affinities. The reason of this "degeneracy" is that the FRET/FLIM measurements provide us with information of the binding affinities, but not of the dynamics of protein complex formation and thus we have no information to constrain the rate at which the protein complexes are formed. To clearly show the logic of our approach we included a panel in Supplementary Figure S5 A to show the

temporal dynamics for the formation of each protein complex using different sets of parameters. These simulations show that while each protein complex reaches a similar steady state (i.e., binding affinities), each parameter combination displays differences in how fast such steady state is reached. As it remains a possibility that some protein complexes are formed faster than others, in the following root stem cell simulations (Supplementary Figure S5 B) our strategy was to select parameter sets randomly and analyse the final steady states of the protein complexes. As pointed out by the reviewer, the different parameters result in a combination of fast and/or slow dynamics that produce variability in the steady state reached for each protein complex in the root stem cells simulations.

We decided to run two control simulations to compare how our root stem cell simulation results change when all parameter describing protein complex formation are either in the fast (highest association parameter predicted) or the slow range (lowest association parameter predicted). As can be seen in Supplementary Figure S8 in both cases the output of the simulations is remarkably similar, thus supporting our approach of focusing on the analysis of the steady states reached.

We have now stated the reason of the degeneracy in the parameter prediction (Methods, lines 842-843) and of the variability of the root stem cell niche simulations (lines 889-890 and 893-896) in our revised manuscript.

Dear Prof. Stahl,

Thank you for the submission of your revised manuscript to our editorial offices. I now looked through the revised manuscript and your further p-b-p-response and consider the remaining points of the referees and the advisor as adequately addressed.

Before we can proceed with formal acceptance, I have these few editorial requests I ask you to address in a final revised manuscript:

- Please remove the information on submission dates, number of words and figures from the manuscript title page.
- Please provide the final manuscript text file without the revision comments.
- Please change the names of the Expanded View figures to Figure EVx in the legend and update their callouts.
- Please add the title of the manuscript to the Appendix title page ('Appendix for ...') and change the names of the Appendix items to "Appendix Figure Sx" or "Appendix Table Sx". Moreover please update all their callouts, also in the Reagents and Tools table.
- Please make sure that all figure panels (main, EV and Appendix figures) are called out separately and sequentially. Presently, there seem to be no callouts for Supplementary Tables S6-S20. Please check. Moreover, please note the point above requesting a name change for these items.
- Please check again that the number "n" for how many independent experiments were performed, their nature (biological versus technical replicates), the bars and error bars (e.g. SEM, SD) and the test used to calculate p-values is indicated in the respective figure legends. Please also check that all the p-values are explained in the legend, and that these fit to those shown in the figure. Please provide statistical testing where applicable. Please avoid the phrase 'independent experiment', but clearly state if these were biological or technical replicates. Please also indicate (e.g. with n.s.) if testing was performed, but the differences are not significant. In case n=2, please show the data as separate datapoints without error bars and statistics. See also:
<http://www.embopress.org/page/journal/14693178/authorguide#statisticalanalysis>

If n<5, please show single datapoints for diagrams. Moreover:

- Please note that the legend for figure 1 is not provided in the sequential manner. This needs to be rectified.
- Please note that the box plots need to be defined in terms of minima, maxima, centre, bounds of box and whiskers, and percentile in the legends of figures 1B, D, F; 3C, 4C, 6B, EV2, EV4 A, B.
- Please note that information related to n is missing in the legend of figure 1H.
- Please note that the measure of center for the error bars needs to be defined in the legend of figure 1H.
- Please note that the scale bar needs to be defined for figures EV1 I, K.
- Please remove the Methods part from the Reagents and Tools Table and provide all Methods information exclusively in the main manuscript text file.
- Please make sure that all the funding information is also entered into the online submission system and that it is complete and similar to the one in the acknowledgement section of the manuscript text file. Presently, grants from the Dutch Research Council (NWO), the Foundation for Food & Agriculture Research (FFAR), companies in the plant breeding and processing industry, and Dutch universities; National Growth Fund (NGF) of the Netherlands; Zeiss LSM 780: DFG- INST 208/551-1 FUGG and Zeiss LSM 880: DFG- INST 208/746-1 FUGG are missing in the submission system. Please check.
- Please use our reference format:
<http://www.embopress.org/page/journal/14693178/authorguide#referencesformat>

Et al needs to be used after 10 author names and DOIs should only be used for preprints and datasets that have not been published yet.

In addition, I would need from you uploaded separately:

- a short, two-sentence summary of the manuscript (not more than 35 words).
- two to four short (!) bullet points highlighting the key findings of your study (two lines each).

Best,

The authors addressed the remaining editorial issues.

Prof. Yvonne Stahl
Goethe University, Frankfurt am Main
Institute for molecular biosciences
Max-von-Laue Str. 9
Frankfurt am Main, Hessen 60438
Germany

Dear Prof. Stahl,

I am very pleased to accept your manuscript for publication in the next available issue of EMBO reports. Thank you for your contribution to our journal.

Yours sincerely,

Rev_Com_number: RC-2024-02499
New_manu_number: EMBOR-2024-60973V3
Corr_author: Stahl
Title: Root stem cell homeostasis in Arabidopsis involves cell type specific transcription factor complexes